

# $x - y$ duality in topological recursion for exponential variables via quantum dilogarithm

Alexander Hock

Mathematical Institute, University of Oxford, Andrew Wiles Building,
Woodstock Road, OX2 6GG, Oxford, UK

alexander.hock@maths.ox.ac.uk

## Abstract

For a given spectral curve, the theory of topological recursion generates two different families $\omega_{g,n}$ and $\omega_{g,n}^{\vee}$ of multi-differentials, which are for algebraic spectral curves related via the universal $x - y$ duality formula. We propose a formalism to extend the validity of the $x - y$ duality formula of topological recursion from algebraic curves to spectral curves with exponential variables of the form $e^x = F(e^y)$ or $e^x = F(y)e^{ay}$ with $F$ rational and $a$ some complex number, which was in principle already observed in [1, 2]. From topological recursion perspective the family $\omega_{g,n}^{\vee}$ would be trivial for these curves. However, we propose changing the $n = 1$ sector of $\omega_{g,n}^{\vee}$ via a version of the Faddeev's quantum dilogarithm which will lead to the correct two families $\omega_{g,n}$ and $\omega_{g,n}^{\vee}$ related by the same $x - y$ duality formula as for algebraic curves. As a consequence, the $x - y$ symplectic transformation formula extends further to important examples governed by topological recursion including, for instance, Gromov-Witten invariants of $\mathbb{C}^3$ (or, equivalently, triple Hodge integrals), orbifold Hurwitz numbers, and stationary Gromov-Witten invariants of $\mathbb{P}^1$. The proposed formalism is related to the issue topological recursion encounters for specific choices of framings for the topological vertex curve.

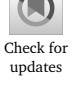

# 1   Introduction

This article mainly deals with the theory of topological recursion (TR) [3] which, roughly speaking, can be seen as an algorithm that associates to a complex curve a family of multi-differentials $\omega_{g,n}$ indexed by $g \in \mathbb{Z}_{\geq 0}$ and $n \in \mathbb{Z}_{>0}$. Depending on the curve these multi-differential can carry information which are of interest in enumerative geometry [4–7], random matrix theory [3, 8–10], topological string theory [11–13], knot theory [14, 15], free probability [16, 17], quantum field theory [18–21], etc. Therefore, TR provides a huge range of applications which goes back to a common algorithm starting from some algebraic curve. The theory of TR was developed around 2007 but it is still a current research topic by itself.

The connection of TR to all these different areas of mathematics and mathematical physics is mostly of a structural nature. It is cumbersome to apply the algorithm of TR to perform explicit computations due to its algorithmic structure, which is recursive in the Euler characteristic $-\chi = 2g + n - 2$. This property prevents direct computations using TR since other techniques are more efficient. For instance, explicit formulas for intersection numbers on the moduli space of complex curves $\overline{\mathcal{M}}_{g,n}$ for small $n$ are well-known and can be derived from Virasoro constraints or localization theory in algebraic geometry. However, TR would require $2g + n - 2$ computational steps.

Changing the roles of $x$ and $y$ in the polynomial, a second family of multi-differentials can be generated by TR, denoted by $\omega_{g,n}^{\vee}$, which clearly distinguishes it from the first family $\omega_{g,n}$. In almost all cases where simple explicit formulas are known for $\omega_{g,n}$ for small $n$, the corresponding dual family $\omega_{g,n}^{\vee}$ is actually trivial, i.e. $\omega_{g,n}^{\vee} = 0$ for $2g + n - 2 > 0$. Therefore, it appears that the existence of simple explicit formulae for the multi-differentials $\omega_{g,n}$ depends on the properties of the dual spectral curve with $x$ and $y$ exchanged. Very recently, completely new insights concerning the relation between $\omega_{g,n}$ and $\omega_{g,n}^{\vee}$ for any algebraic spectral curve were understood; see Fig. 1. Recent works in the context of this duality include [2, 10, 16, 17, 22–26]. Understanding this relation was already considered, for instance, in [27]. The duality between $\omega_{g,n}$ and $\omega_{g,n}^{\vee}$ has resolved, in an extended way, an open problem in the theory of free probability [16], providing the functional relation between the generating series of higher-order free cumulants and moments.

Applying this universal duality formula between $\omega_{g,n}$ and $\omega_{g,n}^{\vee}$ to a curve, which generates a trivial family $\omega_{g,n}^{\vee}$, produces an explicit formula for $\omega_{g,n}$. In other words, for a trivial family $\omega_{g,n}^{\vee}$, we overcome the algorithmic procedure of TR, which is recursive in the Euler characteristic. Consequently, this explains the existence of such explicit formulas, which were generally derived in the past using problem-specific techniques.

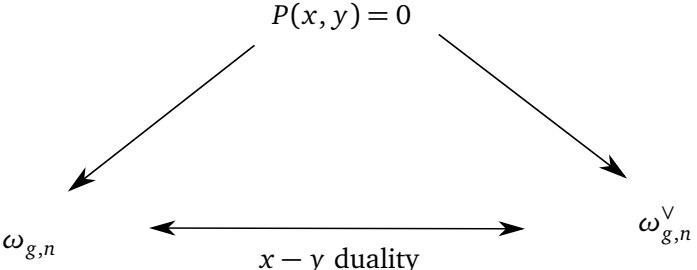

Figure 1: The two families $\omega_{g,n}$ and $\omega_{g,n}^\vee$ which are generated from a curve $P(x,y) = 0$ via TR are related through the universal $x-y$ duality formula.

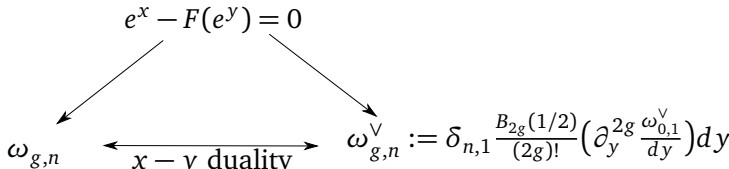

Figure 2: For a curve of the form $e^x - F(e^y) = 0$ (or $e^x - F(y)e^{ay} = 0$) the family $\omega_{g,n}^\vee$ has to be redefined, then the family $\omega_{g,n}$ generated by TR is related to the redefined family $\omega_{g,n}^\vee$ via the universal $x-y$ duality formula.

It is known and observed, for instance, in [26], for the Lambert curve that the $x-y$ duality formula does not hold in general for curves of the form $P(e^x, e^y) = 0$ which have exponential variables. However, very important examples are exactly of this form, such as Gromov-Witten invariants of toric Calabi-Yau 3-folds [11] and conjectured applications in knot theory [15].

This article makes progress in understanding the $x-y$ duality formula for curves of the form $P(e^x, e^y) = e^x - F(e^y) = 0$ (or $e^x - F(y)e^{ay} = 0$) with exponential variables, where $F$ is rational and $a$ some complex number. This family of spectral curves is already a very important subclass with different applications, such as simple Hurwitz numbers (equivalently, linear Hodge integrals) and framed Gromov-Witten invariants of $\mathbb{C}^3$ (equivalently, triple Hodge integrals on $\overline{\mathcal{M}}_{g,n}$).

For curves of this form, the theory of TR would predict a trivial family $\omega_{g,n}^\vee = 0$ for $2g+n-2$. However, we will redefine $\omega_{g,1}^\vee := \frac{B_{2g}(1/2)}{(2g)!}\left(\partial_y^{2g}\frac{\omega_{0,1}^\vee}{dy}\right)dy$ through $\omega_{0,1}^\vee$ [1] and keep $\omega_{g,n}^\vee = 0$ for $n > 1$ and $2g + n - 2 > 0$, see Fig. 2. The coefficient $B_n(x)$ is the Bernoulli polynomial and appears exactly in this form in Faddeev's quantum dilogarithm. It is conjectured that this proposed redefinition of $\omega_{g,n}^\vee$ makes the $x-y$ duality formula work again, which is tested on several examples.[2] The idea of using the form of $\omega_{g,1}^\vee$ comes from the quantum spectral curve $\hat{P}(\hat{x}, \hat{y})$ observed by Gukov and Sulkowski [14], which will be explained along the way in this article.

The paper is organised in the following way:
In Sec. 2, we review facts and definitions about TR, starting in Sec. 2.1 with TR itself. In Sec. 2, we will use a notation with a "tilde" ($\tilde{\omega}_{g,n}$ and $\tilde{\omega}_{g,n}^\vee$) which are the multi-differentials defined by TR directly to distinguish them from the later redefined $\omega_{g,n}^\vee$ related to curves of the form

---

[1]After this article was published, I was informed that for transalgebraic spectral curves the same extension is conjectured [28, Conj. 4.15], where the spectral curve has the form $x(z) = M_0(z)\exp(M_1(z))$, $y(z) = M_2(z)/x(z)$ and $M_i(z)$ meromorphic.

[2]The redefinition and extension proposed in this article were further developed and proved in [29] under the name LogTR. This is now understood as a universal extension of TR for meromorphic $dx$ and $dy$, specifically including logarithmic singularities of $x$ and $y$.

$e^x - F(e^y) = 0$ or $e^x - F(y)e^{ay} = 0$. In Sec. 2.2, we review some background on the quantum spectral curve $\hat{P}(\hat{x}, \hat{y})$ and how to construct the wave function from TR, which is annihilated by the quantum spectral curve. We also discuss the interplay of the $x - y$ duality with the wave function, which is heuristically the Fourier/Laplace transform of the wave function. Then, we recall how the multi-differentials can be reconstructed from the wave function through the kernel $K(x_i, x_j)$ in Sec. 2.3. This reconstruction formula is known as the determinantal formula, which will be later compared with a special case of the $x - y$ duality formula. Then, we review the $x - y$ duality formula in general in Sec. 2.4, which gives the duality between $\tilde{\omega}_{g,n}$ and $\tilde{\omega}_{g,n}^\vee$ for an algebraic curve $P(x, y) = 0$. The special case where $y$ is unramified is discussed separately in Sec. 2.5. We prove, in this case, a new version of the formula which highly resembles the determinantal formula, represented via a summation over permutations $\sigma \in S_n$ consisting just of $n$-cycles.

Sec. 3 considers the curve $e^x - F(e^y) = 0$ (or $e^x - F(y)e^{ay} = 0$), where we use the ordinary notation without "tilde". We start with a motivating example in Sec. 3.1, which has, as a wave function, the Euler $\Gamma$ function. In Sec. 3.2, we explain and motivate the general construction of the two families $\omega_{g,n}$ and $\omega_{g,n}^\vee$ for these specific curves. The family $\omega_{g,n}^\vee$ will be defined by $\omega_{g,n}^\vee = \delta_{n,1} \frac{B_{2g}(1/2)}{(2g)!} \partial_y^{2g} \omega_{0,1}^\vee$, and the family $\omega_{g,n}$ through the $x - y$ duality formula. We propose that for this definition, $\tilde{\omega}_{g,n}$ generated by TR and $\omega_{g,n}$ generated by the $x - y$ formula *coincide*. In Sec. 3.3, we discuss the example of Faddeev's quantum dilogarithm as the wave function. Based on the work of Garoufalidis and Kashaev, we discuss some general constructions for performing Borel resummation for the wave function for curves of the form $e^x - F(e^y) = 0$ in Sec. 3.4. An explicit formula is given for the Borel transform.

In Sec. 4, we apply our proposed construction to curves with important enumerative meaning. We start with the Lambert curve in Sec. 4.1, providing (to our knowledge) new formulas for simple Hurwitz numbers or, equivalently, for linear Hodge integrals (using the ELSV formula). In Sec. 4.2, the framed topological vertex curve is discussed, providing Gromov-Witten invariants of $\mathbb{C}^3$ or equivalently triple Hodge integrals. The last example discusses stationary Gromov-Witten invariants of $\mathbb{P}^1$ where explicit formulas of Pandharipande are reproduced as examples. We also discuss, for this specific curve, that taking limits for the $x - y$ formula is much more convenient than taking limits for TR. This comes from the fact that TR is much more sensitive, where, for instance, ramification points can collide or run to infinity. Therefore, the $x - y$ duality formula seems to be a much more universal duality than just the $x - y$ symplectic transformation in TR.

## 2 Background on topological recursion

This section will provide the necessary background on the theory of topological recursion including the quantum spectral curve, determinantal formula and $x - y$ symplectic duality. We will set the notation used throughout the article focusing on $x - y$ duality in conjunction with the spectral curve and the quantum spectral curve. The explicit result for the $x - y$ duality in case of an algebraic spectral curve will be cited and explained in details to be able to understand the extension to spectral curves of the form $e^x = F(e^y)$ (or $e^x = F(y)e^{ay}$, respectively) in Sec. 3. We will define the corresponding objects in this section with "tilde" (e.g. $\tilde{\omega}_{g,n}$, $\tilde{\omega}_{g,n}^\vee$, $\tilde{W}_{g,n}$, $\tilde{\Phi}_{g,n}$, etc) through the algorithmic definition of topological recursion.

### 2.1 Topological recursion

We understand by the term topological recursion an algorithm which associates to a given initial data, the so-called *spectral curve*, a family of multi-differentials. To be more precise, the

spectral curve is a tuple $(\Sigma, x, y, B)$, where $\Sigma$ is a not necessarily compact Riemann surface, with $x, y : \Sigma \to \mathbb{C}$ are complex functions such that $dx$ and $dy$ are meromorphic, i.e. $x, y$ can have logarithms. Both functions $x, y$ should have at most simple ramification points and log singularities on $\Sigma$, where the ramification points of $x$ are not ramification points of $y$ and vice versa (higher order ramifications are excluded due to technical reasons, see [30] for the definition with higher order ramification points). Then, topological recursion associates to the spectral curve $(\Sigma, x, y, B)$ the multi-differentials $\tilde{\omega}_{g,n}$ living on $\Sigma^n$ with $\tilde{\omega}_{0,1} = y\,dx$ and $\tilde{\omega}_{0,2} = B$, where $B$ is symmetric with double pole on the diagonal and no residue, bi-residue 1 and normalised such that the $A$-periods vanish.

For negative Euler characteristic $\chi = -2g - n + 2 < 0$, all $\tilde{\omega}_{g,n}$ are defined recursively via [3]

$$\tilde{\omega}_{g,n+1}(I,z) := \sum_{\beta_i} \operatorname*{Res}_{q \to \beta_i} K_i(z,q) \Bigg( \tilde{\omega}_{g-1,n+2}(I,q,\sigma_i(q)) + \sum_{\substack{g_1+g_2=g \\ I_1 \uplus I_2 = I \\ (g_i,I_i) \neq (0,\emptyset)}} \tilde{\omega}_{g_1,|I_1|+1}(I_1,q)\tilde{\omega}_{g_2,|I_2|+1}(I_2,\sigma_i(q)) \Bigg).$$

(1)

The following notation is used:

- $I = \{z_1, \ldots, z_n\}$ is a collection of $n$ local coordinates $z_j$ on $\Sigma$, and $I_1, I_2 \subseteq I$ (possibly empty) disjoint subsets of $I$ such that $I_1 \cup I_2 = I$,

- the ramification points $\beta_i$ of $x$ are defined by $dx(\beta_i) = 0$ (or given by poles of $x(z)$ of order greater or equal than 2),

- the local Galois involution $\sigma_i \neq \mathrm{id}$ with $x(q) = x(\sigma_i(q))$ is defined in the vicinity of $\beta_i$ with fixed point $\beta_i$,

- the recursion kernel $K_i(z,q)$ is also locally defined in the vicinity of $\beta_i$ by

$$K_i(z,q) = \frac{\frac{1}{2}\int_{\sigma_i(q)}^{q} B(z,\bullet)}{\tilde{\omega}_{0,1}(q) - \tilde{\omega}_{0,1}(\sigma_i(q))}.$$

Properties which follow (more or less directly) from the definition are

- All $\tilde{\omega}_{g,n}$ are symmetric.

- For $2g + n - 2 > 0$, $\tilde{\omega}_{g,n}$ only has poles located at the ramification points of $x$.

- $\tilde{\omega}_{g,n}$ is homogeneous of degree $2 - 2g - n$, i.e. changing $y \to \lambda y$ or $x \to \lambda x$ by some scalar $\lambda$ transforms $\tilde{\omega}_{g,n} \to \lambda^{2-2g-n}\tilde{\omega}_{g,n}$.

- For $2g + n - 2 > 0$, all $\tilde{\omega}_{g,n}$ are invariant under symplectic transformations of the symplectic form $dx \wedge dy$ which leaves the ramification points $\beta_i$ invariant. These transformations are generated by

$$(x,y) \to \left(\tfrac{ax+b}{cx+d}, \tfrac{(cx+d)^2}{ad-bc}y\right), \quad \text{where} \quad \begin{pmatrix} a & b \\ c & d \end{pmatrix} \in PSL_2(\mathbb{C}),$$

$$(x,y) \to (x, y + R(x)), \quad \text{where} \quad R(x) \text{ is any rational function.}$$

There are several further properties which will not play any role in this article. We refer a curious reader for instance to [3, 31].

Now, we want to fix further notation and definitions for functions used throughout the article. We define

$$\tilde{W}_{g,n}(x(z_1),...,x(z_n))dx(z_1)...dx(z_n) := \tilde{\omega}_{g,n}(z_1,...,z_n),\tag{2}$$

$$\tilde{W}_n(x(z_1),...,x(z_n)) := \sum_{g=0}^{\infty} \hbar^{2g+n-2}\tilde{W}_{g,n}(x(z_1),...,x(z_n)),\tag{3}$$

$$\tilde{\Phi}_{g,n}(x(z_1),...,x(z_n)) := \int^{z_1}...\int^{z_n} \tilde{\omega}_{g,n}(z_1,...,z_n),\tag{4}$$

$$\tilde{\Phi}_n(x(z_1),...,x(z_n)) := \sum_{g=0}^{\infty} \hbar^{2g+n-2}\tilde{\Phi}_{g,n}(x(z_1),...,x(z_n)),\tag{5}$$

where the $\hbar$-series is understood as a formal power series. The integration is just locally defined from some base points close to $z_1,...,z_n$, and the integration constants for $\tilde{\Phi}$ will not play any role. For the sake of this article, only genus zero spectral curves are considered.

The invariance property of the $\tilde{\omega}_{g,n}$ under specific symplectic transformations plays an important role in the theory of TR and also in the theories where TR finds its applications. A third symplectic transformation which transforms the $\tilde{\omega}_{g,n}$'s and does *not* leave them invariant is the $x-y$ symplectic transformation

$$(x,y) \mapsto (y,x).\tag{6}$$

Strictly speaking, this is a symplectic transformation up to a sign $-1$ which can be restored by the homogeneity property. To clearly distinguish between $\tilde{\omega}_{g,n}$ generated by the spectral curve $(\Sigma, x, y, B)$ or by the curve $(\Sigma, y, x, B)$, we will use the notation $\vee$ as superscript. Therefore, we define the $\tilde{\omega}_{g,n}^{\vee}$ to be generated by the spectral curve $(\Sigma, y, x, B)$ by the same TR algorithm (1) but with $x$ and $y$ interchanged. For instance, we have $\tilde{\omega}_{0,1}^{\vee} = x\,dy$ and $\tilde{\omega}_{0,2}^{\vee} = B = \tilde{\omega}_{0,2}$.

The families $\tilde{\omega}_{g,n}$ and $\tilde{\omega}_{g,n}^{\vee}$ of multi-differentials are *dual* to each other. If the family $\tilde{\omega}_{g,n}$ can be reconstructed from the family $\tilde{\omega}_{g,n}^{\vee}$ through a specific formula, then the family $\tilde{\omega}_{g,n}^{\vee}$ can also be reconstructed from the family $\tilde{\omega}_{g,n}$ via the same formula but $x$ and $y$ interchanged.

Similar to the previous definitions (with integration again defined locally and just for a genus zero spectral curve), we will use throughout the article the following functions

$$\tilde{W}_{g,n}^{\vee}(y(z_1),...,y(z_n))dy(z_1)...dy(z_n) := \tilde{\omega}_{g,n}^{\vee}(z_1,...,z_n),\tag{7}$$

$$\tilde{W}_n^{\vee}(y(z_1),...,y(z_n)) := \sum_{g=0}^{\infty} \hbar^{2g+n-2}\tilde{W}_{g,n}^{\vee}(y(z_1),...,y(z_n)),\tag{8}$$

$$\tilde{\Phi}_{g,n}^{\vee}(y(z_1),...,y(z_n)) := \int^{z_1}...\int^{z_n} \tilde{\omega}_{g,n}^{\vee}(z_1,...,z_n),\tag{9}$$

$$\tilde{\Phi}_n^{\vee}(y(z_1),...,y(z_n)) := \sum_{g=0}^{\infty} \hbar^{2g+n-2}\tilde{\Phi}_{g,n}^{\vee}(y(z_1),...,y(z_n)).\tag{10}$$

The relation between all $\tilde{W}_{g,n}$ and $\tilde{W}_{g,n}^{\vee}$ will be presented in Sec. 2.4 for spectral curves with meromorphic $x, y : \Sigma \to \mathbb{C}$ on a compact Riemann surface $\Sigma$, i.e. there exists an irreducible polynomial $P(x(z), y(z)) = 0$ for $z \in \Sigma$. However, the relation for the first few examples can be read off by the definitions. For $(g,n) = (0,1)$, we have

$$\tilde{W}_{0,1}(x(z)) = y(z), \qquad \tilde{W}_{0,1}^{\vee}(y(z)) = x(z).$$

It implies that $\tilde{W}_{0,1}$ is the formal inverse of $\tilde{W}_{0,1}^\vee$, i.e. $\tilde{W}_{0,1}(\tilde{W}_{0,1}^\vee(y(z))) = y(z)$ and $\tilde{W}_{0,1}^\vee(\tilde{W}_{0,1}(x(z))) = x(z)$. For $(g, n) = (0, 2)$, we have

$$\tilde{W}_{0,2}(x(z_1), x(z_2)) = \frac{B(z_1, z_2)}{dx(z_1)dx(z_2)}, \qquad \tilde{W}_{0,2}^\vee(y(z_1), y(z_2)) = \frac{B(z_1, z_2)}{dy(z_1)dy(z_2)},$$

from which one concludes

$$\tilde{W}_{0,2}(x(z_1), x(z_2)) = \tilde{W}_{0,2}^\vee(y(z_1), y(z_2)) \frac{dy(z_1)}{dx(z_1)} \frac{dy(z_2)}{dx(z_2)}.$$

Just as a side remark, the functional relation for $\tilde{W}_{0,2}$ is the second order moment-cumulant functional relation in the theory of free probability, see [16,17,32] for more information about the relation to the theory of free probability.

The relation between $\tilde{W}_{g,n}$ and $\tilde{W}_{g,n}^\vee$ can thus be interpreted as a generalization of an inversion formula graded by the Euler characteristic $2g + n - 2$ of the two integers $g$ and $n$. In the case of $g = 0$ and $n = 1$, this simply corresponds to the classical inversion of a function $\tilde{W}_{0,1}^\vee(y) = x(y)$ and $\tilde{W}_{0,1}(x) = y(x)$, respectively.

## 2.2 Quantum spectral curve

The quantum spectral curve is an important application of TR to ordinary differential equations and its solutions. The idea of quantum spectral curve from TR could be traced back to [33]. It was suggested for the $A$-polynomial by Gukov and Sulkowski [14]. For an accessible review on quantum spectral curve see [34]. The WKB solution is reconstructed from TR as explained later, this is proved for genus zero curves in [35]. For higher genus algebraic curves, new insight was gained in [36] and extended to hyperelliptic curves in [37, 38] and further generalised in [39]. The perturbative WKB solution has to be extended to a non-perturbative wave function. However, we will just review some facts about the WKB solution via TR (genus zero spectral curves), and refer the reader to the literature mentioned above for higher genus spectral curves.

A spectral curve $(\Sigma, x, y, B)$ with rational $x, y : \Sigma \to \mathbb{C}$ and compact $\Sigma$ can also be represented as the vanishing locus of a polynomial $P(x, y) = 0$ as mentioned before. Being more precise, we have a complex curve of genus zero defined by

$$\{P(x(z), y(z)) = 0 \,|\, \forall z \in \Sigma\}.$$

Now, we are interested in an operator-valued quantisation of the polynomial in a quantum mechanical sense. The polynomial $P$ is quantised with $x \to \hat{x} = x$ and $y \to \hat{y} = \hbar \frac{d}{dx}$ to $\hat{P}(\hat{x}, \hat{y})$, together with the semi-classical limit $\lim_{\hbar \to 0} \hat{P}(x, y) = P(x, y)$. The quantisation of $P$ to $\hat{P}$ is obviously not unique due to ambiguities by ordering the operators $\hat{x}$ and $\hat{y}$ and additional $\hbar$ terms which vanish in the semi-classical limit. Nevertheless, having such a differential operator $\hat{P}$ one might ask for the wave function $\tilde{\Psi}(x)$ which is annihilated by it, i.e.

$$\hat{P}(\hat{x}, \hat{y})\tilde{\Psi}(x) = 0. \tag{11}$$

The algorithm of TR provides a way to compute a wave function $\tilde{\Psi}(x)$. Let $(\Sigma, x, y, B)$ be a given spectral curve of genus zero represented as $P(x, y) = 0$, then there *exists* a quantisation $\hat{P}(\hat{x}, \hat{y})$ which annihilates the (perturbative) wave function

$$\tilde{\Psi}(x) := \exp\left( \sum_{g \geq 0, n \geq 1} \frac{\hbar^{2g+n-2}}{n!} \tilde{\Phi}_{g,n}(x, ..., x) \right), \tag{12}$$

where $\tilde{\Phi}_{g,n}$ is defined in (4), and the special case $\tilde{\Phi}_{0,2}$ is regularised by $\int \int \tilde{\omega}_{0,2}(z_1, z_2) - \frac{dx(z_1)\,dx(z_2)}{(x(z_1)-x(z_2))^2}$. Here, the integration constants of $\tilde{\Phi}$ actually play some role, but we do not go into the details. The explicit form of the quantum spectral curve can be construct from the structure of the singularities (see [35]). For higher genus algebraic spectral curve, the construction of the wave function from TR includes non-perturbative parts, and is much more involved [39].

In the special case, where the polynomial is not algebraic but takes the form $e^x = F(e^y)$ (or $e^x = F(y)e^{ay}$, respectively), where $F$ is rational, a quantum spectral curve was conjectured in [14, eq. (3.20)] which annihilates the wave function constructed by TR via (12) to be of the form

$$\hat{P}(e^{\hat{x}}, e^{\hat{y}}) = e^{\hat{x}+\frac{\hbar}{2}} - F\left(e^{\hat{y}-\frac{\hbar}{2}}\right). \tag{13}$$

However, it was already observed in [14, 40] that there are some issues if the spectral curve $e^x = F(e^y)$ does not have ramification points in $x$, which includes the framed topological vertex curve with framing $f = 0$. These observations and comments in the literature seem not to be resolved in the meantime, but they are highly related to the construction in Sec. 3 in conjunction with the $x - y$ duality.

The quantum spectral curve and the wave function can of course also be studied from the $x - y$ duality perspective This means there is a second quantum spectral curve of the form $\hat{P}^{\vee}(\hat{x}, \hat{y})$ with operators $\hat{x} = \hbar\frac{d}{dy}$ and $\hat{y} = y$ and the semi-classical limit $\lim_{\hbar \to 0}\hat{P}^{\vee}(x, y) = P(x, y)$. The dual wave function $\tilde{\Psi}^{\vee}(y)$ is constructed again via TR but with interchanged roles of $x$ and $y$, i.e.

$$\tilde{\Psi}^{\vee}(y) := \exp\left(\sum_{g \geq 0, n \geq 1} \frac{\hbar^{2g+n-2}}{n!}\tilde{\Phi}_{g,n}^{\vee}(y, ..., y)\right), \tag{14}$$

where $\tilde{\Phi}_{g,n}^{\vee}$ is defined in (9), and the special case $\tilde{\Phi}_{0,2}^{\vee}$ is regularised by $\int \int \tilde{\omega}_{0,2}^{\vee}(z_1, z_2) - \frac{dy(z_1)\,dy(z_2)}{(y(z_1)-y(z_2))^2}$.

So what is now the relation between the two wave function $\tilde{\Psi}(x)$ and $\tilde{\Psi}^{\vee}(y)$? The naive suggestion would be of course that both wave functions are related via *Fourier/Laplace transformation*, which from another perspective implies that the $x - y$ symplectic transformation in TR is deeply related to a Fourier/Laplace transformation.

Let us give two examples for the quantum spectral curve, the $x - y$ duality and its interrelation, where the first one is almost too simple:

**Example 2.1.** *Take the trivial spectral curve* $(\mathbb{C}, x = z, y = z, \frac{dz_1\,dz_2}{(z_1-z_2)^2})$ *which is represented by the polynomial* $P(x, y) = x - y = 0$. *For this example, $x$ and $y$ have no ramification points which implies that all $\tilde{\omega}_{g,n}$ and $\tilde{\omega}_{g,n}^{\vee}$ vanish for $2g + n - 2 > 0$. However, $\tilde{\omega}_{0,1}$ and $\tilde{\omega}_{0,1}^{\vee}$ are not trivial. Just applying the previous definitions, we find*

$$\tilde{\Phi}_{0,1}(x) = \frac{x^2}{2} \quad \rightarrow \quad \tilde{\Psi}(x) = \exp\left(\frac{x^2}{2\hbar}\right),$$

$$\tilde{\Phi}_{0,1}^{\vee}(y) = \frac{y^2}{2} \quad \rightarrow \quad \tilde{\Psi}^{\vee}(y) = \exp\left(\frac{y^2}{2\hbar}\right).$$

*Both wave functions are the Gaussian function, where it is very well known that its Fourier transform is again a Gaussian function. The wave functions are annihilated by the quantum spectral curves $\hat{P}(\hat{x}, \hat{y}) = x - \hbar\frac{d}{dx}$ and $\hat{P}^{\vee}(\hat{x}^{\vee}, \hat{y}^{\vee}) = \hbar\frac{d}{dy} - y$, respectively.*

**Example 2.2.** *Take the simplest nontrivial example, the Airy spectral curve* $(\mathbb{C}, x = z^2, y = z, \frac{dz_1\,dz_2}{(z_1-z_2)^2})$ *which is represented by the polynomial* $P(x, y) = x - y^2 = 0$. *In*

this example, $y$ has no ramification points implying that all $\tilde{\omega}_{g,n}^{\vee} = 0$ with $2g + n - 2 > 0$. From the previous definitions, we find

$$\tilde{\Phi}_{0,1}^{\vee}(y) = \frac{y^3}{3} \quad \rightarrow \quad \tilde{\Psi}^{\vee}(y) = \exp\left(\frac{y^3}{3\hbar}\right),$$

satisfying the differential equation

$$\left(\hbar \frac{d}{dy} - y^2\right)\tilde{\Psi}^{\vee}(y) = 0.$$

On the other hand, all $\tilde{\omega}_{g,n}$ do not vanish since $x$ has a ramification point at $z = 0$. It is very well-known that $\tilde{\omega}_{g,n}$ computes the $\psi$-class intersection numbers on $\overline{\mathcal{M}}_{g,n}$ the moduli space of complex curves with $n$ marked points [3], which was essentially observed by Kontsevich [41] in proving Witten's conjecture [42]. Furthermore, the wave function generated by these $\tilde{\omega}_{g,n}$'s gives actually the asymptotic expansion of the Airy function (or Bairy function, depending on the sign of $\hbar$) [35, 43]

$$\tilde{\Psi}_{\pm}(x) = \frac{e^{\mp\frac{2}{3\hbar}x^{3/2}}}{\sqrt{2\pi}x^{1/4}} \sum_{k=0}^{\infty} \frac{(6k)!}{1296^k(2k)!(3k)!}\left(\mp\frac{\hbar}{x^{3/2}}\right)^k,$$

where $\tilde{\Psi}_{+}(x)$ corresponds to the Airy function and $\tilde{\Psi}_{-}(x)$ to the Bairy function. The wave functions satisfy

$$\left(\left(\hbar\frac{d}{dx}\right)^2 - x\right)\tilde{\Psi}_{\pm}(x) = 0.$$

The Airy (or Bairy function, respectively) can be obtained by a Fourier/Laplace transform of the dual wave function $\tilde{\Psi}^{\vee}(y)$, where the contour depends on the complex argument of $\hbar$

$$\tilde{\Psi}_{\pm}(x) = \frac{1}{\sqrt{2\pi\hbar}} \int_{C_{\pm}} e^{xy/\hbar}\tilde{\Psi}^{\vee}(y)dy.$$

## 2.3 Reconstruction of $\tilde{\omega}_{g,n}$ from the wave function

The reconstruction of $\tilde{\omega}_{g,n}$ or equivalently $\tilde{W}_{g,n}$ is given by the determinantal formula [33] through the kernel $K(x_1, x_2)$, which by itself is constructed from all wave functions (this means from all linearly independent solutions of the differential equation (11))

$$K(x_1, x_2) = \frac{\exp\left(\sum_{n=1}^{\infty} \frac{\int_{x_2}^{x_1} \dots \int_{x_2}^{x_1} \tilde{W}_n(x_1', \dots, x_n')dx_1' \dots dx_n'}{n!}\right)}{x_1 - x_2}, \tag{15}$$

where one has to be very careful since the pullback to the $x$-space from the $z$-space chooses a specific branch, or equivalently a specific solution of the quantum spectral curve. The determinantal formula is therefore another way of representing the formal power series $\tilde{W}_n = \sum_g \hbar^{2g+n-2}\tilde{W}_{g,n}$ by an explicit formula if the kernel is known. The determinantal formula was recently used in combination with resurgence to derive the large genus asymptotitcs for intersection numbers on $\overline{\mathcal{M}}_{g,n}$ at subleading order [43].

**Example 2.3.** *The Airy kernel $K_{Airy}$ is constructed by the Airy function $\tilde{\Psi}_{+}(x)$ and the B-Airy function $\tilde{\Psi}_{-}(x)$ of Example 2.2 which are the two independent solutions of the differential equation $\hbar^2\tilde{\Psi}''(x) - x\tilde{\Psi} = 0$:*

$$K_{Airy}(x_1, x_2) = \frac{1}{\hbar}\frac{\tilde{\Psi}_{+}(x_1)\tilde{\Psi}_{-}'(x_2) - \tilde{\Psi}_{+}'(x_1)\tilde{\Psi}_{-}(x_2)}{x_1 - x_2}.$$

Another way is to construct the kernel from its differential system (see [33] for details), but this is beyond the scope of the article.

We want to emphasise the structure of the determinantal formula, and compare the different structures of the determinantal formula and the $x - y$ duality formula (18). Let $K(x_1, x_2)$ be the kernel (15) (also used in [24, eq. (3.8)]), then it was conjectured in [33] that the correlators

$$\tilde{W}_n(x_1, ..., x_n) = (-1)^{n-1} \sum_{\substack{\sigma \in S_n \\ \sigma = n\text{-cycle}}} \prod_{i=1}^{n} K(x_i, x_{\sigma(i)}) - \frac{\delta_{n,2}}{(x_1 - x_2)^2}, \tag{16}$$

$$\tilde{W}_1(x) = \lim_{x' \to x} \left( K(x, x') - \frac{1}{x - x'} \right), \tag{17}$$

are equal to $\tilde{W}_n$ generated by TR (1). The sum over all permutations $\sigma \in S_n$ is restricted to permutations with one cycle of length $n$.

## 2.4  $x - y$ duality for algebraic curves

Now, we want to recall the relation between $\tilde{\omega}_{g,n}$ generated by the spectral curve $(\Sigma, x, y, B)$ and $\tilde{\omega}_{g,n}^{\vee}$ generated by the spectral curve $(\Sigma, y, x, B)$ with interchanged $x$ and $y$. We assume $x, y$ to be meromorphic and have simple distinct ramification points. It turns out that it is much more convenient to represent this functional relation in terms of $\tilde{W}_n$ and $\tilde{W}_n^{\vee}$ defined in (3) and (8).

We use in this article the $x - y$ formula as it appeared in [17, 23] or in [25] for the special case of genus $g = 0$. For this, we will need first to define the following graphs:

**Definition 2.4.** *Let $\mathcal{G}_n$ be the set of connected bicoloured graphs $\Gamma$ with $\bigcirc$-vertices and $\bullet$-vertices, where the number of $\bigcirc$-vertices is $n$. A graph $\Gamma$ satisfies the following conditions:*

- *the $\bigcirc$-vertices are labelled from $1, ..., n$,*

- *edges are only connecting $\bullet$-vertices with $\bigcirc$-vertices,*

- *$\bullet$-vertices have valence $\geq 2$.*

*For a graph $\Gamma \in \mathcal{G}_n$, let $r_i(\Gamma)$ be the valence of the $i^{th}$ $\bigcirc$-vertex.*

*Let $I \subset \{1, ..., n\}$ be the set associated to a $\bullet$-vertex, where $I$ is the set of labels of $\bigcirc$-vertices connected to this $\bullet$-vertex. Let $\mathcal{I}(\Gamma)$ be the set of all sets $I$ for a given graph $\Gamma \in \mathcal{G}_n$.*

We will abuse the notation $x_i = x(z_i) = x_i(z_i)$ and $y_i = y(z_i) = y_i(z_i)$ for convenience. This implies by chain rule that $x_i$ depends on $y_i$ and vice versa. Thus, we understand $\frac{dy_i}{dx_i} = \frac{dy(z_i)}{dx(z_i)} = \frac{1}{x'(z_i)} \frac{dy(z_i)}{dz_i} = \frac{y'(z_i)}{x'(z_i)}$. Then, the functional relation between the two families $\tilde{W}_n$ and $\tilde{W}_n^{\vee}$ is the following sum over graphs (see [17, 23] for more details and examples):

**Theorem 2.5.** *Let $(\Sigma, x, y, B)$ be an algebraic spectral curve, and let $\tilde{W}_n^{\vee}(y_1, ..., y_n) := \sum_{g=0}^{\infty} \hbar^{2g+n-2} \tilde{W}_{g,n}^{\vee}(y_1, ..., y_n)$ be generated by TR on the spectral curve $(\Sigma, y, x, B)$. Let further be $S(u) = \frac{e^{u/2} - e^{-u/2}}{u}$ and for $I = \{i_1, ..., i_n\}$*

$$\hat{c}^{\vee}(u_I, y_I) := \left( \prod_{i \in I} \hbar u_i S(\hbar u_i \partial_{y_i}) \right) \left( \tilde{W}_n^{\vee}(y_I) \right),$$

*and for $I = \{j, j\}$ the special case*

$$\hat{c}^{\vee}(u_I, y_I) := (\hbar u_j S(\hbar u_j \partial_{y_j}))(\hbar u_j S(\hbar u_j \partial_y)) \left( \tilde{W}_2^{\vee}(y_j, y) - \frac{1}{(y_j - y)^2} \right) \Bigg|_{y = y_j}.$$

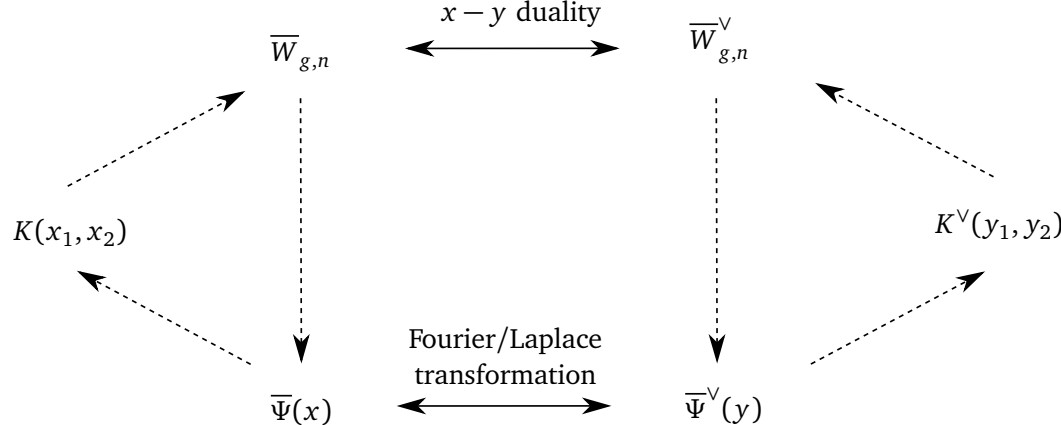

Figure 3: Duality interplay.

*Define the following differential operator acting from the left*

$$\hat{O}^\vee(y_i) := \sum_{m \geq 0} \left(-\partial_{x_i}\right)^m \left(-\frac{dy_i}{dx_i}\right)[u_i^m] \frac{\exp\left(\hbar u_i S(\hbar u_i \partial_{y_i}) \tilde{W}_1^\vee(y_i) - x_i u_i\right)}{\hbar u_i}.$$

*Then, all $\tilde{W}_{g,n}$ generated by TR by the spectral curve $(\Sigma, x, y, B)$ are related to the $\tilde{W}_{g',n'}^\vee$ via formal power series in $\hbar$ of*

$$\tilde{W}_n(x_1, ..., x_n) = \sum_{\Gamma \in \mathcal{G}_n} \frac{1}{|\mathrm{Aut}(\Gamma)|} \prod_{i=1}^n \hat{O}^\vee(x_i) \prod_{I \in \mathcal{I}(\Gamma)} \hat{c}^\vee(u_I, y_I), \tag{18}$$

*where the graphs $\mathcal{G}_n$ are defined in Definition 2.4.*

This theorem is proved in [23] and provides a solution for an open problem in the theory of TR. Actually, the theorem gives a duality between two families of correlators $\tilde{W}_{g,n}$ and $\tilde{W}_{g,n}^\vee$, since the same formula holds by interchanging $x$ and $y$ as well as $\tilde{W}_{g,n}$ with $\tilde{W}_{g,n}^\vee$. The functional relation of the theorem has solved simultaneously an open problem in the theory of free probability [32] related higher order free cumulants and moments [16]. In the same vein, it relates from a combinatorial perspective ordinary maps and fully simple maps [9,10].

Remember that the family $\tilde{W}_{g,n}$ gives rise to the wave function $\tilde{\Psi}(x)$ and the dual family $\tilde{W}_{g,n}^\vee$ to the wave function $\tilde{\Psi}^\vee(y)$ via (12) and (14), respectively. Since both wave functions $\tilde{\Psi}(x)$ and $\tilde{\Psi}^\vee(y)$ are related via Fourier/Laplace transformation, we can conclude that the functional relation of Theorem 2.5 is the corresponding duality formula for each $g$ and $n$. Therefore, the $x - y$ duality formula (18) is, casually speaking, a way to generate asymptotically a Fourier/Laplace transformation which is graded by the Euler characteristic. At each grading the functions $\tilde{W}_{g,n}^\vee$ are functions in several variables. Fig. 3 shows the interplay between the different dualities, the Fourier/Laplace transformation between the wave function $\tilde{\Psi}$ and $\tilde{\Psi}^\vee$ as well as the $x - y$ duality between $\tilde{W}_{g,n}$ and $\tilde{W}_{g,n}^\vee$. The dashed lines show the reconstructions mentioned before. The remaining duality between the two kernels $K$ and $K^\vee$ was recently considered in [24] for KP integrable systems.

## 2.5   $x - y$ duality for algebraic curves with unramified $y$

The formula becomes particularly nice and applicable if one of the families, for instance $\tilde{W}_{g,n}^\vee$, is trivial. An important subclass of genus zero spectral curves are curves of the form $x = F(y)$,

where $F$ is rational which implies that $y$ is unramified. Therefore, the family $\tilde{W}_{g,n}^{\vee}$ vanishes identically. In this case Theorem 2.5 breaks down to:

**Corollary 2.6.** *Consider the situation of a genus zero spectral curve with simple ramification points for $x$ and $y$ is unramified, i.e. $dy(z) \neq 0$ for all $z \in \Sigma$. In particular, we can set $y(z) = z$ or $y(z) = \frac{1}{z}$. Then, all $\tilde{W}_{g,n}^{\vee} = 0$ for $2g + n - 2 > 0$, and therefore all $\hat{c}^{\vee}(u_I, y_I) = 0$ for $|I| > 2$. In this specific case, the $x - y$ formula reduces to (see [23, 26] for further details)*

$$\tilde{W}_n(x_1(z_1), ..., x_1(z_n)) = \prod_{i=1}^{n} \hat{O}^{\vee}(y_i(z_i)) \sum_{\Gamma \in \mathcal{G}_n^2} \prod_{\{i,j\} \in \mathcal{I}(\Gamma)} \frac{\hbar^2 u_i u_j}{(y_i(z_i) - y_i(z_j))^2 - \frac{\hbar^2}{4}(u_i + u_j)^2}, \quad (19)$$

*where $\mathcal{G}_n^2 \subset \mathcal{G}_n$ is the set of graphs defined in Definition 2.4 with only 2-valent •-vertices just connecting two different ◯-vertices and at most one •-vertex connects the same pair of ◯-vertices, or equivalently (and much simpler) $\mathcal{G}_n^2$ is the set of connected labeled graphs with $n$ vertices*[3]

With the formula (19) of Corollary 2.6, one can compute all $\tilde{W}_{g,n}$ for any $g$ directly. It is not necessary any more to follow the algorithmic procedure of TR recursively in the Euler characteristic. Examples which are included in the class of spectral curves considered in Corollary 2.6 are the enumeration of $\psi$-class intersection numbers [41, 42], $r$-spin intersection numbers [7, 44], $\Theta$-class intersection numbers and $r$-spin $\Theta$-class intersection numbers on $\tilde{\mathcal{M}}_{g,n}$ [45, 46] or the enumeration of genus permutations [47].

Interestingly, the $x - y$ formula of Corollary 2.6 can be brought into a formula of exactly the same shape as the determinantal formula (16) for algebraic curves with unramified $y$. The sum over all connected labeled graphs in (19) can indeed be turned into a sum over permutations $\sigma \in S_n$ where $\sigma$ is an $n$-cycle. As a consequence we get another explicit formula with significantly less terms. The number of terms is reduced since the number of $n$-cycle permutations in $S_n$ is $(n-1)!$, whereas the number of connected graphs with $n$ labeled vertices grow much faster. Actually, the asymptotic growth of connected graphs with $n$ labeled vertices is $2^{\binom{n}{2}} = 2^{\frac{n(n-1)}{2}}$ since almost all graphs with $n$ labeled vertices are connected (see [48, p. 138]).

**Proposition 2.7.** *Consider the situation of a genus zero spectral curve with simple ramification points for $x$ and $y$ is unramified, i.e. $dy(z) \neq 0$ for all $z \in \Sigma$. In particular, we can set $y(z) = z$ or $y(z) = \frac{1}{z}$. Then, all $\tilde{W}_{g,n}$ generated by TR from this spectral curve are*

$$\tilde{W}_{g,n}(x_1(z_2), ..., x_n(z_n)) = [\hbar^{2g+n-2}] \prod_{i=1}^{n} \hat{O}^{\vee}(y_i(z_i)) \hbar u_i \sum_{\substack{\sigma \in S_n \\ \sigma = n\text{-cycle}}} \prod_{i=1}^{n} \frac{1}{y_i(z_i) + \frac{\hbar u_i}{2} - y_{\sigma(i)}(z_{\sigma(i)}) + \frac{\hbar u_{\sigma(i)}}{2}},$$
$$(20)$$

*and in the special case $n = 1$, the sum over all $n$-cycles is defined to be $\frac{1}{\hbar u_1}$.*

*Proof.* The aim is to transform the rhs of (19) into the rhs of (20). The first step is to rewrite

$$\frac{\hbar^2 u_i u_j}{(y_i - y_j)^2 - \frac{\hbar^2}{4}(u_i + u_j)^2} = \frac{(y_i - y_j)^2 - \frac{\hbar^2}{4}(u_i - u_j)^2}{(y_i - y_j)^2 - \frac{\hbar^2}{4}(u_i + u_j)^2} - 1,$$

where the subtraction of 1 corresponds to just taking connected correlators which was in principle already used to derive Corollary 2.6 (see [23, 26]).

Then, we go from connected correlators $\tilde{W}_{g,n}(x_1, ..., x_n)$ to disconnected $\overset{\circ}{\tilde{W}}_{g,n}(x_1, ..., x_n)$ defined as sum over all partitions

$$\overset{\circ}{\tilde{W}}_n(x_I) := \sum_{\lambda \vdash I} \prod_{i=1}^{l(\lambda)} \tilde{W}_{|\lambda_i|}(x_{\lambda_i}). \quad (21)$$

---

[3]See for instance A001187 on OEIS, where the number of connected labeled graphs with $n$ vertices is listed.

We achieve an equation for the disconnected $\mathring{\tilde{W}}_{g,n}$, where the sum over all connected graphs turns into the product

$$\mathring{\tilde{W}}_n(x_1(z_1),...,x_1(z_n)) = \prod_{i=1}^{n} \hat{O}^{\vee}(y_i(z_i)) \prod_{1 \le i < j \le 1} \frac{(y_i - y_j)^2 - \frac{\hbar^2}{4}(u_i - u_j)^2}{(y_i - y_j)^2 - \frac{\hbar^2}{4}(u_i + u_j)^2}. \tag{22}$$

Combining the factor $\frac{1}{\hbar u_i}$ from $\hat{O}^{\vee}(y_i(z_i))$ and the product $\prod_{1 \le i < j \le n}$, we can write it in the form of the determinant of the Cauchy matrix

$$\prod_{i=1}^{n} \frac{1}{a_i + b_i} \prod_{i<j} \frac{(a_i - b_j)(a_j - b_i)}{(a_i + b_j)(a_j - b_i)} = \det\left(\frac{1}{a_i + b_j}\right) = \sum_{\sigma \in S_n} \text{sign}(\sigma) \prod_i \frac{1}{a_i + b_{\sigma(i)}}, \tag{23}$$

with $a_i = -y_i + \frac{\hbar u_i}{2}$ and $b_j = y_j + \frac{\hbar u_j}{2}$. Restricting to the connected part, which is nothing than the Möbius function for the partially ordered set of partitions of $\{1,...,n\}$, all terms coming from permutations $\sigma \in S_n$ with more than one cycle vanish. The remaining $n$-cycle permutations $\sigma \in S_n$ have parity $n-1$ which gives an overall factor of $\text{sign}(\sigma) = (-1)^{n-1}$. This proves the assertion. $\qquad\square$

Applying this formula to some known interesting example like the Airy curve, $r$-spin Airy curve and Lambert curve will be provided:

**Example 2.8.** *For the Airy curve* $(\mathbb{C}, x = z^2, y = z, \frac{dz_1\,dz_2}{(z_1-z_2)^2})$, *Proposition 2.7 yields the following representation for* $\tilde{W}_{g,n}$ *which are the same as computed by TR:*

$$\tilde{W}_{g,n}(x_1(z_1),...,x_n(z_n)) \tag{24}$$

$$= [\hbar^{2g+n-2}] \prod_{i=1}^{n} \left( \sum_{m_i} \left(-\frac{\partial}{\partial(z_i)^2}\right)^{m_i} [u_i^{m_i}] \frac{e^{\frac{\hbar^2 u_i^3}{12}}}{2z_i} \right) \sum_{\substack{\sigma \in S_n \\ \sigma = n\text{-cycle}}} \prod_{i=1}^{n} \frac{1}{z_i + \frac{\hbar u_i}{2} - z_{\sigma(i)} + \frac{\hbar u_{\sigma(i)}}{2}}.$$

*The* $\psi$-*class intersection numbers are extracted from* $\tilde{W}_{g,n}$ *via*

$$\tilde{W}_{g,n}(x_1(z_1),...,x_n(z_n)) = \sum_{k_1,...,k_n=3g+n-3} \langle \psi_1^{k_1}...\psi_n^{k_n} \rangle_{g,n} \prod_{i=1}^{n} \frac{(2k_i+1)!!}{2z_i^{2k_i+3}}. \tag{25}$$

*An explicit formula for the intersection numbers can be extracted by taking the Laplace transformation of* (25) *with* $\frac{1}{\sqrt{2\pi}} \int_{-\infty}^{\infty} dx(z_i) \frac{e^{-\mu_i z_i^2}}{z_i^{2k_i+3}} = \frac{\mu_i^{k_i+1/2}}{(2k_i+1)!!}$ *for the rhs, which can further be represented as a geometric series. For the lhs, we insert* (24) *and perform integration by parts* $m_i$ *times (see a detailed discussion for this in [26]) to finally get*

$$\left\langle \prod_{i=1}^{n} \frac{\sqrt{\mu_i}}{1 - \mu_i \psi_i} \right\rangle_{g,n} = [\hbar^{2g+n-2}] \prod_{i=1}^{n} \left( \frac{e^{\frac{\hbar^2 \mu_i^3}{12}}}{\sqrt{2\pi}} \int_{-\infty}^{\infty} dz_i e^{-\mu_i z^2} \right) \sum_{\substack{\sigma \in S_n \\ \sigma = n\text{-cycle}}} \prod_{i=1}^{n} \frac{1}{z_i + \frac{\hbar \mu_i}{2} - z_{\sigma(i)} + \frac{\hbar \mu_{\sigma(i)}}{2}},$$

*where the contours are on the real line with a small semi circle above the origin. This formula coincides with the formula which can be found in [49, 50].*

**Example 2.9.** *The* $r$-*spin Ariy spectral curve is a generalisation of the form* $(\mathbb{C}, x = z^r, y = z, \frac{dz_1\,dz_2}{(z_1-z_2)^2})$. *This spectral curve has for positive integer* $r$ *greater than 2 higher order ramification. The corresponding version of TR which keeps track of higher order ramification was formulated in [30]. A new cohomological class* $c_W(a_1,...,a_n) \in H^{\bullet}(\overline{\mathcal{M}}_{g,n}, \mathbb{Q})$ *of degree*

$s = \frac{(r-2)(g-1)+\sum_i(a_i-1)}{r}$ (which has to be a positive integer $s \in \mathbb{N}$) was defined by Witten in [51] for the moduli space complex curves associated with an $r$-spin structure. A conjecture of Witten concerning the intersection number $\langle \tau_{k_1,a_1}...\tau_{k_n,a_n}\rangle_{g,n} = \langle c_W(a_1,...,a_n)\psi_1^{k_1}...\psi_n^{k_n}\rangle_{g,n}$ of this class $c_W(a_1,...,a_n)$ with $\psi$-classes relates to the $r$-KdV hierarchy, which was proved [52]. We use the notation of [43], where the relation of these intersection numbers to TR and the determinantal formula was discussed. The correlators $\tilde{W}_{g,n}$ computed by higher order TR are related to the intersection numbers via

$$\tilde{W}_{g,n}(x_1(z_1),...,x_n(z_n)) = \sum_{\substack{0 \le k_1,...,k_n \\ 1 \le a_1,...,a_n \le r-1}} \prod_{i=1}^n (-r)^{g-1-|k|} \langle \tau_{k_1,a_1}...\tau_{k_n,a_n}\rangle_{g,n} \prod_{i=1}^n \frac{(rk_i+a_i)!_{(r)}}{rz_i^{r(k_i+1)+a_i}}, \quad (26)$$

where $m!_{(r)}$ is the $r$-factorial defined recursively for $m > r$ by $m!_{(r)} = m(m-r)!_{(r)}$ and $m!_{(r)} = m$ for $0 < m \le r$.

On the other hand, assuming Proposition 2.7 is true for TR with higher order ramification (which is just conjectured at the moment, see also [23, Remark 5.8]) yields the following representation for $\tilde{W}_{g,n}$ (note that applying the higher order generalisation of TR from [30] is quite tedious to use)

$$\tilde{W}_{g,n}(x_1(z_1),...,x_n(z_n)) = [\hbar^{2g+n-2}] \prod_{i=1}^n \left( \sum_{m_i} \left( -\frac{\partial}{\partial(z_i)^r} \right)^{m_i} [u_i^{m_i}] \frac{e^{\frac{(z_i+\frac{\hbar u_i}{2})^{r+1}-(z_i-\frac{\hbar u_i}{2})^{r+1}}{\hbar(r+1)} - z_i^r u_i}}{rz_i^{r-1}} \right)$$

$$\times \sum_{\substack{\sigma \in S_n \\ \sigma = n\text{-cycle}}} \prod_{i=1}^n \frac{1}{z_i + \frac{\hbar u_i}{2} - z_{\sigma(i)} + \frac{\hbar u_{\sigma(i)}}{2}}. \quad (27)$$

Again, taking the Laplace transformation of (26) with $\frac{1}{\Gamma(1-\frac{a_i}{r})} \int_0^\infty dx_i \frac{e^{-\mu_i x_i}}{x_i^{k_i+a_i/r+1}} = (-1)^{k_i} \frac{r^{k_i}\mu_i^{k_i+a_i/r}}{(rk_i+a_i)!_{(r)}}$ for each $i$, we extract the intersection number. The lhs behaves nicely under the Laplace transformation as described in [26] such that we conclude

$$\sum_{1 \le a_1,...,a_n \le r-1} \left\langle c_W(a_1,...,a_n)\prod_{i=1}^n \frac{\Gamma(1-\frac{a_i}{r})\mu_i^{a_i/r}}{1-\mu_i\psi_i} \right\rangle_{g,n} (-r)^{g-1}$$

$$= [\hbar^{2g+n-2}] \prod_{i=1}^n \left( \int_0^\infty dz_i e^{-\frac{(z_i+\frac{\hbar\mu_i}{2})^{r+1}-(z_i-\frac{\hbar\mu_i}{2})^{r+1}}{\hbar(r+1)}} \right) \sum_{\substack{\sigma \in S_n \\ \sigma = n\text{-cycle}}} \prod_{i=1}^n \frac{1}{z_i + \frac{\hbar\mu_i}{2} - z_{\sigma(i)} + \frac{\hbar\mu_{\sigma(i)}}{2}}. \quad (28)$$

This formula recovers the formula of Brezin and Hikami [44,53], which can be used in the $n = 1$ case to derive more explicit formulas [54,55].

Note that the formula (28) depends on the rhs continuously on $r$, whereas using higher order TR [30] to derive explicit values for these intersection numbers needs a fixed integer $r$ corresponding to the order of ramification. When $n > 1$, the integration contours are shifted slightly relative to each other.

**Example 2.10.** *The Lambert spectral curve [4,56] computes simple Hurwitz numbers which due to the ELSV formula to linear Hodge integrals on $\overline{\mathcal{M}}_{g,n}$. The spectral curve takes the form $(\mathbb{C}, x = z - \log z, y = \log z, \frac{dz_1 dz_2}{(z_1-z_2)^2})$, where $y$ was shifted by $x$ to make the $x - y$ formula applicable, see [26] for details. Let $c_k(\mathbb{E})$ be the $k$th Chern class of the Hodge bundle $\mathbb{E}$ and $\Lambda(\alpha) = 1 + \sum_{k=1}^g (-1)^k \alpha^{-k} c_k(\mathbb{E})$. Then, the correlators generated by TR compute the intersection numbers of the Hodge classes in the following way*

$$\tilde{W}_{g,n}(x_1(z_1),...,x_n(z_n)) = \sum_{k_1,...,k_n \ge 0} \prod_{i=1}^n \frac{k_i^{k_i+1}}{k_i!} \left\langle \frac{\Lambda(1)}{\prod_{i=1}^n(1-k_i\psi_i)} \right\rangle_{g,n} e^{k_i x_i(z_i)}. \quad (29)$$

*Since $y = \log z$, we can not use the formula from Corollary 2.6 and Proposition 2.7. However assuming that Thm 2.5 holds for this curve,[4] computations analogous to Corollary 2.6 and the proof of Proposition 2.7 with $y = \log z$ provide the formula (Cauchy determinant has to be used with $a_i = z_i e^{\hbar k_i/2}$ and $b_j = -z_j e^{-\hbar k_i/2}$)*

$$\tilde{W}_{g,n}(x_1(z_1), ..., x_n(z_n)) \tag{30}$$

$$= [\hbar^{2g+n-2}] \prod_{i=1}^{n} \left( \sum_{m_i} \left( -\frac{\partial}{\partial x_i(z_i)} \right)^{m_i} [u_i^{m_i}] \frac{e^{u_i(S(\hbar u_i)-1)z_i}}{1-z_i} \right) \sum_{\substack{\sigma \in S_n \\ \sigma = n\text{-cycle}}} \prod_{i=1}^{n} \frac{1}{z_i e^{\frac{\hbar k_i}{2}} - z_{\sigma(i)} e^{-\frac{\hbar k_{\sigma(i)}}{2}}} .$$

*Taking now the Laplace transformation of (29) with contour around the origin by $\mathrm{Res}_{z_i} e^{(u_i - \mu_i)x_i(z_i)} dx_i(z_i) = \delta_{\mu_i, u_i}$ extracts the intersection number. On the other hand, applying this Laplace transformation to (30) as described in [26] yields finally*

$$\prod_{i=1}^{n} \frac{k_i^{k_i+1}}{k_i!} \left\langle \frac{\Lambda(1)}{\prod_{i=1}^{n}(1-k_i\psi_i)} \right\rangle_{g,n} = \mathrm{Res}_{z_i=0}[\hbar^{2g+n-2}] \prod_{i=1}^{n} \frac{dz_i e^{k_i z_i S(\hbar k_i)}}{z_i^{k_i}} \sum_{\substack{\sigma \in S_n \\ \sigma = n\text{-cycle}}} \prod_{i=1}^{n} \frac{1}{z_i e^{\frac{\hbar k_i}{2}} - z_{\sigma(i)} e^{-\frac{\hbar k_{\sigma(i)}}{2}}} .$$

*The difference to the formula already appearing in [26] is that the sum is over n-cycle permutations rather than over connected labelled graphs.*

# 3 $x - y$ duality for curves of the form $e^x = F(e^y)$ or $e^x = F(y)e^{ay}$

It was already observed in [14, 40] that TR possess some issues with curves of the form $e^y = F(e^x)$ which are for instance the framed topological vertex curve with framing $f = 0$. To understand on a very simple example that the $x - y$ symplectic transformation formula is directly related to this observation, we will present an almost trivial example related to the Euler $\Gamma$-function as wave function of the quantum spectral curve. This example carries already all information needed to understand the general formalism which makes the $x - y$ formula applicable to the curves under consideration.

In this section we switch to a notation without "tilde", and will explain in Sec. (3.2) how these objects are defined in detail. Assume $\omega_{g,n}$ and $\omega_{g,n}^{\vee}$ are given (the explicit definition of these objects will be given in Sec. (3.2)), then the relation to $W_{g,n}$, $W_n$, $W_{g,n}^{\vee}$, $W_n^{\vee}$, $\Phi_{g,n}$, $\Phi_n$, $\Phi_{g,n}^{\vee}$, $\Psi(x)$ and $\Psi^{\vee}(y)$ is given as before (2)-(5), (7)-(10), (12) and (14) by just removing "tilde".

## 3.1 The $\Gamma$-wave function

Just for this subsection (due to pedagogical reasons), we consider the spectral curve $(\mathbb{P}^1, x, y, B)$ of the form $e^y = ...$ (rather than $e^x = ...$) with

$$e^y = x, \tag{31}$$

with the parametrisation

$$x = z, \qquad y = \log z.$$

Quantising naively this spectral curve via $x \to \hat{x} = x$ and $y \to \hat{y} = \hbar \frac{\partial}{\partial x}$ leads to the formal differential operator $e^{\hbar \frac{\partial}{\partial x}} - x$ which annihilates formally the Euler $\Gamma$ function

$$(e^{\hbar \frac{\partial}{\partial x}} - x)\Gamma\left(\frac{x}{\hbar}\right) = 0,$$

---

[4]At the time this article was written, this was just a conjecture which was proved later after submission in [29].

where we have used the action $e^{\hbar \frac{\partial}{\partial x}} f(x) = f(x + \hbar)$ and $\Gamma(1+x) = x\Gamma(x)$. Note this is just a formal observation and all functions are understood in terms of formal expansions in $\hbar$. The asymptotic[5] (not convergent) expansion of the logarithm of the $\Gamma$ function at infinity is very well-known in terms of Bernoulli numbers

$$\log \Gamma\left(\frac{x}{\hbar}\right) \sim \frac{x}{\hbar} \log \frac{x}{\hbar} - \frac{x}{\hbar} - \frac{1}{2} \log \frac{x}{\hbar} + \frac{1}{2} \log 2\pi + \sum_{k=2}^{\infty} \frac{B_k \hbar^{k-1}}{k(k-1)x^{k-1}}.$$

Comparing this to the construction of the wave function $\tilde{\Psi}(x)$ via $\tilde{\omega}_{g,n}$ as defined in Sec. 2.2, we find heuristic arguments that the $\tilde{\omega}_{g,n}$ should actually not vanish for $2g + n - 2 > 0$ if the wave function is constructed as suggested in Sec. 2.2 . However, applying TR to the curve defined in (31) has vanishing $\tilde{\omega}_{g,n} = 0$ for all $2g + n - 2 > 0$, since $x(z) = z$ is unramified, i.e. $dx(z) \neq 0$ for all $z \in \mathbb{P}^1$.

Having the $x - y$ symplectic transformation in mind, we apply TR for the curve with $x$ and $y$ exchanged. In this situation, we find also vanishing $\tilde{\omega}_{g,n}^{\vee} = 0$ for all $2g + n - 2 > 0$ since $y(z) = \log z$ has no algebraic ramification point. After having swapped $x$ and $y$, consider heuristically the quantum spectral curve with the quantisation $x \to \hat{x} = \hbar \frac{\partial}{\partial y}$ and $y \to \hat{y} = y$. Direct computation shows

$$\left(e^y - \hbar \frac{\partial}{\partial y}\right) e^{\frac{e^y}{\hbar}} = 0,$$

where the logarithms of the wave function has no contributions at order $\hbar^n$ and $n \geq 0$, which aligns with the observation that all $\tilde{\omega}_{g,n}^{\vee} = 0$ for $2g + n - 2 > 0$.

We conclude that there is an obvious mismatch by naively trying to construct the wave function from the correlators $\tilde{\omega}_{g,n}$ generated by TR from the curve (31). However, we show that the $x - y$ symplectic transformation formula resolves this problem in the following way.

Now we change the notation to $\omega_{g,n}$ without "tilde". Consider the spectral curve (31) with $\omega_{g,n}^{\vee} \equiv 0$ for $2g + n - 2 > 0$, then we *define* all $\omega_{g,n}$ through the $x - y$ symplectic transformation formula (18) by replacing $\tilde{\omega}_{g,n}^{\vee}$ by $\omega_{g,n}^{\vee}$, which means replacing $\tilde{W}_{g,n}^{\vee}$ by $W_{g,n}^{\vee}$. The first examples already shows that $W_{g,n} = \frac{\omega_{g,n}}{dx_1 \ldots dx_n}$ does not vanish for $2g + n - 2 > 0$ with $n = 1$. It turns out that the corresponding wave function constructed as in Sec. 2.2 from this $\omega_{g,n}$ is the $\Gamma$-function. More precisely, we have

$$\Psi(x) = \frac{\Gamma(\frac{x}{\hbar} + \frac{1}{2}) \cdot \hbar^{\frac{x}{\hbar}}}{\sqrt{2\pi}},$$

which is annihilated by the operator

$$\hat{P}(x, \hbar \partial_x) = e^{\hbar \partial_x} - x - \frac{\hbar}{2}, \tag{32}$$

which is in accordance with the conjecture by Gukov and Sulkowski (13).

To see this, take $x = z$ and $y = \log z$. We assume that all $W_{g,n}^{\vee} = 0$ for $2g + n - 2 > 0$ since $y$ is unramified. Consider the cases $(g, n) = (g, 1)$. Apply the $x - y$ duality formula of Corollary 2.6 for $W_{g,1}$

$$W_{g,1}(x) = [\hbar^{2g-1}] \sum_k (-\partial_x)^k [u^k] \frac{\exp(xu(S(\hbar u) - 1))}{\hbar u S(\hbar u)} \frac{(-1)}{x},$$

---

[5]Note an asymptotic expansion at $x = 0$ for the form $f(x) \sim \sum_n a_n x^n$ satisfies $|f(x) - \sum_{n=0}^{N} a_n x^n| \in \mathcal{O}(x^{N+1})$ where $a_n$ can have factorial growth.

where $S(x) = \frac{e^{x/2}-e^{-x/2}}{x} = 1 + \frac{x^2}{24} + ....$ Integrating once with respect to $x$ gives $\Phi_{g,1}(x) = \int W_{g,1}(x)dx$ which changes just a factor of $u$. We have as a formal power series

$$\Phi_{g,1}(x) = [\hbar^{2g-1}]\sum_k (-\partial_x)^k[u^k]\frac{\exp(xu(S(\hbar u)-1))}{\hbar u^2 S(\hbar u)}\frac{(-1)}{x}.$$

Note expanding the exponential in $\hbar$ has a higher power of $u$ than of $x$. This means that after acting with $(-\partial_x)^k[u^k]$ on it, this term vanishes. In summary, the exponential does only contribute as a 1. We find

$$\begin{aligned}
\Phi_{g,1}(x) &= [\hbar^{2g-1}]\sum_k (-\partial_x)^k[u^k]\frac{1}{\hbar u^2 S(\hbar u)}\frac{(-1)}{x}\\
&= [\hbar^{2g-1}]\sum_k (-\partial_x)^k[u^k]\frac{1}{\hbar S(\hbar u)}(x - x\log x)\\
&= [\hbar^{2g-1}]\frac{e^{\hbar/2\partial_x}\hbar\partial_x}{\hbar(e^{\hbar\partial_x}-1)}(x - x\log x)\\
&= [\hbar^{2g-1}]e^{\hbar/2\partial_x}\left(x/\hbar\log x/\hbar - x/\hbar - \frac{1}{2}\log x/\hbar + \sum_{k=2}^{\infty}\frac{B_k}{k(k-1)(x/\hbar)^{k-1}}\right) + \frac{x}{\hbar}\log\hbar\\
&= [\hbar^{2g-1}]\log\Gamma\left(\frac{x}{\hbar}+\frac{1}{2}\right) + \frac{x}{\hbar}\log\hbar - \frac{1}{2}\log 2\pi,
\end{aligned}$$

which coincides with the $\Gamma$ function. Next, we argue that all $W_{g,n}$ with $n > 2$ and $2g+n-2 > 0$ vanish. The poles at the diagonal vanish due to the argument in [23, Proposition 4.6] (nothing changes by including logarithms). The only possible pole would be at $x_i = 0$ which is generated by the term $\frac{dy_i}{dx_i}$, however the expansion of the exponential yields factors of $x_i$ together with factors of $u_i^{2+n}$. These generate higher derivatives with respect to $x_i$. So, we use the same argument as before which means we have terms of the form $\partial_{x_i}^{n+k}x_i^n$, where $k > 0$. An explicit computation is very cumbersome and there is no new insight.

In conclusion, the wave function is constructed by just $W_{g,1}(x)$

$$\log\Psi(x) = \sum_{g=0}^{\infty}\hbar^{2g-1}\Phi_{g,1}(x) = \log\Gamma\left(\frac{x}{\hbar}+\frac{1}{2}\right) + \frac{x}{\hbar}\log\hbar - \frac{1}{2}\log 2\pi.$$

Via the functional equation of the $\Gamma$ function, the quantum spectral curve (32) is also derived.

**Remark 3.1.** *The dual wave function is*

$$\Psi^{\vee}(y) = \exp\left(\sum_{g,n}\frac{\hbar^{2g+n-2}}{n!}\Phi^{\vee}_{g,n}(y,...,y)\right) = \exp\left(\frac{e^y}{\hbar}-\frac{1}{2}y\right),$$

*since $\Phi^{\vee}_{0,2}(y,y) = -y$. It is annihilated by $\hat{P}^{\vee}(\hbar\partial_y,y) = e^y - \hbar\partial_y - \frac{\hbar}{2}$. Its formal Fourier/Laplace transform with an appropriate contour and $\hbar$ chosen in an appropriate region in the complex plane is equal to the wave function $\Psi(x)$*

$$\frac{1}{\sqrt{\hbar 2\pi}}\int \underbrace{e^{\left(\frac{e^y}{\hbar}-\frac{1}{2}y\right)}}_{=\Psi^{\vee}(y)}e^{-xy/\hbar}dy = \frac{1}{\sqrt{\hbar 2\pi}}\int e^{-t}t^{x/\hbar-1/2}\hbar^{x/\hbar+1/2}dt = \underbrace{\frac{\Gamma(\frac{x}{\hbar}+\frac{1}{2})\hbar^{x/\hbar}}{\sqrt{2\pi}}}_{=\Psi(x)},$$

*where we have substituted $\frac{e^y}{\hbar} = -t$.*

We summarise the previous observation in the following way. TR implies that both families $\tilde{\omega}_{g,n}$ and $\tilde{\omega}_{g,n}^{\vee}$ vanish for the spectral curve (31). However, constructing the wave functions from these trivial families $\tilde{\omega}_{g,n}$ and $\tilde{\omega}_{g,n}^{\vee}$ gives wrong results. Now, assuming that the family $\omega_{g,n}^{\vee} = \tilde{\omega}_{g,n}^{\vee}$ is trivial and letting $\omega_{g,n}$ be defined by the $x - y$ duality formula produces a non-vanishing family $\omega_{g,n}$ which generates the correct wave function. Therefore, one of the families should actually not be trivial even though TR implies it.

## 3.2  General construction

We propose the following extension of the $x - y$ formula for spectral curves of the form $e^x = F(e^y)$ (or $e^x = F(y)e^{ay}$) with exponential variables having, a priori from TR perspective, one trivial family of correlators $\tilde{\omega}_{g,n}^{\vee} = 0$. The formalism extends the trivial family $\tilde{\omega}_{g,n}^{\vee}$ to $\omega_{g,n}^{\vee}$ by a version of the asymptotic series of Faddeev's quantum dilogarithm [57]. The connection to Faddeev's quantum dilogarithm will be explained in the next subsection.

We define for the spectral curve $e^x = F(e^y)$ (or $e^x = F(y)e^{ay}$) the correlators

$$W_{g,1}^{\vee}(y) := \frac{B_{2g}(1/2)}{(2g)!} \partial_y^{2g}(W_{0,1}^{\vee}(y)), \tag{33}$$

where $W_{0,1}(y) = x(y) = \log F(e^y)$. The coefficient $B_n(x)$ is the $n$th Bernoulli polynomial, and more precisely the coefficient appearing (33) is the $2g$ coefficient of the function $\frac{1}{S(x)} = \frac{x}{e^{x/2} - e^{-x/2}}$, i.e. $[x^{2g}]\frac{x}{e^{x/2} - e^{-x/2}} = \frac{B_{2g}(1/2)}{(2g)!}$. Note that this coefficient already appeared in the expansion of the $\Gamma$ function before.

Now we *define* all $W_{g,n}$ through the $x - y$ symplectic transformation formula (18) which are in general nontrivial with poles at the ramification points of $x$ for $2g + n - 2 > 0$.

Now, we have the following proposal (which is conjectural an checked for many examples):

> If we have a spectral curve of the form $e^x = F(e^y)$ (or $e^x = F(y)e^{ay}$) with $F$ rational and simple algebraic ramification points for $x$, then the correlators $\tilde{\omega}_{g,n}$ computed by TR via (1) coincide with the correlators $\omega_{g,n}$ from the $x - y$ formula (18) where $W_{g,1}^{\vee}$ is defined by (33), i.e.
>
> $$\tilde{\omega}_{g,n} = \omega_{g,n}.$$

**Remark 3.2.** *When this paper was finished, I was informed by Alexander Alexandrov, Boris Bychkov, Petr Dunin-Barkowski, Maxim Kazarian, and Sergey Shadrin that this proposal is in some sense a special case of the results in [1, 2]. However, it was not formulated in this way directly in the past. An equivalent version of the explicit formulas (34) or (35) were instead directly derived and proved to satisfy TR, and not taken as a consequence of the proposed extension of the $x - y$ duality formula. Therefore, this paper still provides a new perspective on the topic revolving around the $x - y$ duality in general.*

Considering (33), we find, as a consequence of the $x - y$ duality formula, Corollary 2.6, and Proposition 2.7, the explicit form

$$
W_n(x_1, ..., x_n) = \prod_{i=1}^{n} \sum_{m_i \geq 0} \left(-\frac{\partial}{\partial x_i}\right)\left(-\frac{\partial y_i}{\partial x_i}\right)[u_i^{m_i}]
$$
$$
\times \exp\left(\sum_{g=0}^{\infty} \hbar^{2g-1}(\Phi_{g,1}^{\vee}(y_i + \hbar u_i/2) - \Phi_{g,1}^{\vee}(y_i - \hbar u_i/2)) - x_i u_i + y_i\right)
$$
$$
\times \sum_{\substack{\sigma \in S_n \\ \sigma = n\text{-cycle}}} \prod_{i=1}^{n} \frac{1}{e^{y_i + \frac{\hbar k_i}{2}} - e^{y_{\sigma(i)} - \frac{\hbar k_{\sigma(i)}}{2}}}, \tag{34}
$$

for curves of the form $e^{x_i} = F(e^{y_i})$.

In case of curves of the form $e^{x_i} = F(y_i)e^{ay_i}$, we have the minor change

$$
W_n(x_1, ..., x_n) = \prod_{i=1}^{n} \sum_{m_i \geq 0} \left(-\frac{\partial}{\partial x_i}\right)\left(-\frac{\partial y_i}{\partial x_i}\right)[u_i^{m_i}]
$$
$$
\times \exp\left(\sum_{g=0}^{\infty} \hbar^{2g-1}(\Phi_{g,1}^{\vee}(y_i + \hbar u_i/2) - \Phi_{g,1}^{\vee}(y_i - \hbar u_i/2)) - x_i u_i\right)
$$
$$
\times \sum_{\substack{\sigma \in S_n \\ \sigma = n\text{-cycle}}} \prod_{i=1}^{n} \frac{1}{y_i + \frac{\hbar k_i}{2} - y_{\sigma(i)} + \frac{\hbar k_{\sigma(i)}}{2}}, \tag{35}
$$

where $\Phi_{0,1}^{\vee}(y) = \int x \, dy = \int \log F(y) dy$ and $\Phi_{g,1}^{\vee}(y) = \frac{B_{2g}(1/2)}{(2g)!} \partial_y^{2g}(\Phi_{0,1}^{\vee}(y))$.

Note that the argument of the exponential in both expressions has the following equivalent representation:

$$
\sum_{g=0}^{\infty} \hbar^{2g-1}(\Phi_{g,1}^{\vee}(y + \hbar u/2) - \Phi_{g,1}^{\vee}(y - \hbar u/2)) - xu = \left(\frac{S(u\hbar\partial_y)}{S(\hbar\partial_y)} - 1\right)u\, x(y),
$$

where we remind $S(x) = \frac{e^{x/2} - e^{-x/2}}{x}$. The form is very similar to the original formulation of the $x - y$ duality formula appearing in [2, 58].

Let us justify the choice of (33) with a prediction for the quantum spectral curve of the form $e^x = F(e^y)$ by Gukov and Sułkowski [14, (3.20)], who conjecture

$$
\hat{P}^{\vee}(\hat{x}, \hat{y}) = e^{\hat{x} - \frac{\hbar}{2}} - F\left(e^{\hat{y} + \frac{\hbar}{2}}\right),
$$

where the operators are $\hat{x} = \hbar\partial_y$ and $\hat{y} = y$. The corresponding wave function $\Psi^{\vee}(y)$ will therefore satisfy the functional relation

$$
\Psi^{\vee}(y + \hbar)e^{-\frac{\hbar}{2}} - F\left(e^{y + \frac{\hbar}{2}}\right)\Psi^{\vee}(y) = 0,
$$

or equivalently (after shifting $y \to y - \frac{\hbar}{2}$ and renormalising the wave function $\Psi^{\vee, ren}(y) = \Psi^{\vee}(y)e^{-y/2}$, which is essentially subtracting the contribution of $\Phi_{0,2}^{\vee}$)

$$
\Psi^{\vee, ren}\left(y + \frac{\hbar}{2}\right) - F(e^y)\Psi^{\vee, ren}\left(y - \frac{\hbar}{2}\right) = 0.
$$

Now, it becomes obvious that the logarithm of the renormalised wave function together with the action of the formal derivative $e^{\frac{\hbar}{2}\partial_y} - e^{\frac{\hbar}{2}\partial_y}$ satisfies

$$
(e^{\frac{\hbar}{2}\partial_y} - e^{-\frac{\hbar}{2}\partial_y})\log\Psi^{\vee, ren}(y) = \log\Psi^{\vee, ren}\left(y + \frac{\hbar}{2}\right) - \log\Psi^{\vee, ren}\left(y - \frac{\hbar}{2}\right) = \log F(e^y).
$$

Inverting formally the operator $(e^{\frac{\hbar}{2}\partial_y} - e^{\frac{\hbar}{2}\partial_y})$ and adding on the rhs $1 = \partial_y \partial_y^{-1}$ yields

$$\log \Psi^{\vee,ren}(y) = \frac{1}{\hbar S(\hbar \partial_y)} \int \log F(e^y) dy = \frac{1}{\hbar S(\hbar \partial_y)} \Phi_{0,1}^{\vee}(y), \tag{36}$$

where $S(t) = \frac{e^{t/2} - e^{-t/2}}{t}$. Expanding both sides in a formal power series in $\hbar$, we identify the terms of order $\hbar^{2g-1}$ in the lhs (which is $\Phi_{g,1}^{\vee}(y)$) with $\frac{B_{2g}(1/2)}{(2g)!} \partial_y^{2g}(\Phi_{0,1}^{\vee}(y))$ from the rhs, under the assumption that all $\Phi_{g,n}^{\vee}(y)$ with $n > 1$ vanish (except $\Phi_{0,2}^{\vee}(y)$ which contributed to the renormalisation).

## 3.3 Faddeev's quantum dilogarithm and $x - y$ duality

The quantum dilogarithm [57] (see also [59–61] and references therein for further information) is a special function which plays an important role in quantum Teichmüller theory and complex Chern-Simons theory. The quantum dilogarithm is defined by

$$\tilde{\varphi}_b(x) = \exp\left(\int_{\mathbb{R}+i\varepsilon} \frac{e^{-2ixt}}{4\sinh(tb)\sinh(t/b)} \frac{dt}{t}\right).$$

The Fourier transform is given by (see for instance [62, eq. (A.16)])

$$\int_{-\infty}^{\infty} \tilde{\varphi}_b(x) e^{2\pi i yx} dx = \frac{e^{-\frac{\pi i}{12}(3+b^2+b^{-2})-\pi y(b+b^{-1})}}{\tilde{\varphi}_b\left(-y - \frac{i}{2}(b + b^{-1})\right)}. \tag{37}$$

For our purpose, it is convenient to use the following representation after variable transformation

$$\varphi_\hbar(x) := \tilde{\varphi}_{\sqrt{\frac{\hbar}{2\pi i}}}(x/\sqrt{-\hbar 2\pi i}) = \exp\left(\int \frac{e^{xt}}{S(t\hbar)(e^{i\pi t} - e^{-i\pi t})} \frac{dt}{\hbar t^2}\right), \tag{38}$$

which satisfies $\varphi_\hbar(x) = \frac{1}{\varphi_{-\hbar}(x)}$ and the functional relation

$$\varphi_\hbar\left(x - \frac{\hbar}{2}\right) = (1 + e^x)\varphi_\hbar\left(x + \frac{\hbar}{2}\right), \tag{39}$$

and has the asymptotic expansion

$$\log \varphi_\hbar(x) \sim \sum_{g=0}^{\infty} \hbar^{2g-1} \frac{B_{2g}(1/2)}{(2g)!} \partial_x^{2g} \text{Li}_2(-e^x) = \frac{1}{\hbar S(\hbar \partial_x)} \text{Li}_2(-e^x), \tag{40}$$

which is exactly of the same form as the new defined family of $W_{g,1}^{\vee}$ with $W_{0,1}^{\vee}(y) = \log(-1 - e^y)$.

Thus to make the connection to TR, we want to study the spectral curve

$$e^x + e^y + 1 = 0, \tag{41}$$

which is symmetric between $x$ and $y$. This curve is very special, since the curve can be written as $e^x = F(e^y)$ and $e^y = F(e^x)$. The general construction from Sec. 3.2 tells us that both $W_{g,1}$

and $W_{g,1}^\vee$ should not vanish and actually be

$$\Phi_{0,1}(x) = \int y \, dx = x\pi i - \text{Li}_2(-e^x), \tag{42}$$

$$\Phi_{g,1}(x) = \frac{B_{2g}(1/2)}{(2g)!} \partial_x^{2g}(\Phi_{0,1}(x)), \tag{43}$$

$$\Phi_{0,1}^\vee(y) = \int x \, dy = y i\pi - \text{Li}_2(e^{-y}), \tag{44}$$

$$\Phi_{g,1}^\vee(y) = \frac{B_{2g}(1/2)}{(2g)!} \partial_y^{2g}(\Phi_{0,1}^\vee(y)). \tag{45}$$

Now, we want to check if this aligns with the proposed $x - y$ duality formula. Therefore, take (34) for $n = 1$ which is

$$W_{g,1}(x) = [\hbar^{2g-1}] \sum_{m \geq 0} \left(-\frac{\partial}{\partial x}\right)^m \left(-\frac{\partial y}{\partial x}\right)$$
$$\times [u^m] \frac{\exp\left(\sum_{k=0}^\infty \hbar^{2k-1}(\Phi_{k,1}^\vee(y + \hbar u/2) - \Phi_{k,1}^\vee(y - \hbar u/2)) - xu\right)}{\hbar u S(\hbar u)}, \tag{46}$$

where $\Phi_{g,1}^\vee$ is given by (45). Comparing now the lhs with $W_{g,1}(x)$ of (44), we are expecting that the rhs computes to $\frac{B_{2g}(1/2)}{(2g)!} \partial_x^{2g}(W_{0,1}(x))$. This is already generated if we set the exponential to be 1, since

$$[\hbar^{2g-1}] \sum_{m \geq 0} \left(-\frac{\partial}{\partial x}\right)^m \left(-\frac{\partial y}{\partial x}\right)[u^m]\frac{1}{\hbar u S(\hbar u)} = \frac{B_{2g}(1/2)}{(2g)!} \partial_x^{2g+1}(\Phi_{0,1}(x)) = \frac{B_{2g}(1/2)}{(2g)!} \partial_x^{2g}(W_{0,1}(x)).$$

Next, we multiply (46) by $\hbar^{2g-1}$ and sum over $g$. Acting with $e^{\hbar \partial_x/2} - e^{-\hbar \partial_x/2}$ on both sides generates for the lhs (if $W_{g,1}(x) = \partial_x \Phi_{g,1}(x)$ of (43))

$$\sum_{g=0}^\infty \hbar^{2g-1}(W_{g,1}(x + \hbar/2) - W_{g,1}(x - \hbar/2)) = \partial_x \frac{e^{\hbar\partial_x/2} - e^{-\hbar\partial_x/2}}{e^{\hbar\partial_x/2} - e^{-\hbar\partial_x/2}} W_{0,1}(x) = \frac{\partial y}{\partial x}.$$

Whereas on the rhs the action of $e^{\hbar\partial_x/2} - e^{-\hbar\partial_x/2}$ cancels against the $S(\hbar u)$ in the denominator and we obtain the following statement:

$$-\frac{\partial y}{\partial x} = [\hbar^{2g}] \sum_{m \geq 0} \left(-\frac{\partial}{\partial x}\right)^m \left(-\frac{\partial y}{\partial x}\right)[u^m] \exp\left(\sum_{k=0}^\infty \hbar^{2k-1}(\Phi_{k,1}^\vee(y + \hbar u/2) - \Phi_{k,1}^\vee(y - \hbar u/2)) - xu\right),$$

with $e^x + e^y + 1 = 0$ and $\Phi_{g,1}^\vee$ given by (45). This means that *every order $\hbar^{2g}$ with $g > 1$ vanish identically*. This was tested with computer algebra system up to genus $g = 5$, and it seems to be completely non trivial.

One can summarise the previous computation in the following statement:
*The $x - y$ duality formula sends coefficients of the quantum dilogarithm to coefficients of the quantum dilogarithm in a nontrivial way.*

Next, we discuss compatibility with the quantum spectral curve. Both wave functions $\Psi(x)$ and $\Psi^\vee(y)$ can directly be constructed from $\Phi_{g,1}(x)$ and $\Phi_{g,1}^\vee(y)$ since all other $\Phi_{g,n}^{(\vee)} = 0$ for

$n \geq 2$ except for $\Phi_{0,2}^{(\vee)}$. We deduce

$$\log \Psi(x) = \frac{1}{2}x + \frac{i\pi}{2} + \frac{x\pi i}{\hbar} + \sum_{g=0}^{\infty}(-\hbar)^{2g-1}\frac{B_{2g}(1/2)}{(2g)!}\partial_x^{2g}(\text{Li}_2(-e^x)),$$

$$\log \Psi^{\vee}(y) = \frac{1}{2}y + \frac{i\pi}{2} + \frac{y\pi i}{\hbar} + \sum_{g=0}^{\infty}(-\hbar)^{2g-1}\frac{B_{2g}(1/2)}{(2g)!}\partial_y^{2g}(\text{Li}_2(-e^y)),$$

where the first two terms for each wave function come from $\Phi_{0,2}^{(\vee)}$. Due to the functional relation of the quantum dilogarithm $\varphi(x)$ (39) and $\Psi(x) = \varphi_{-\hbar}(x)e^{\frac{x}{2}+\frac{x\pi i}{\hbar}}$, we find that $\Psi(x)$ satisfies

$$(1 + e^{x+\hbar/2})\Psi(x) + \Psi(x+\hbar)e^{-\hbar/2} = 0,$$

implying that the quantum spectral curve of (41) reads

$$\hat{P}(\hat{x},\hat{y}) = 1 + e^{\hat{x}+\hbar/2} + e^{\hat{y}-\hbar/2}. \tag{47}$$

This obviously aligns with the prediction of Gukov and Sulkowski (13). It is straightforward to check that the dual wave function $\Psi^{\vee}(y) = \varphi_{\hbar}(y)e^{\frac{y}{2}+\frac{y\pi i}{\hbar}}$ satisfies

$$(1 + e^{y-\hbar/2})\Psi^{\vee}(y) + \Psi^{\vee}(y-\hbar)e^{\hbar/2} = 0,$$

where the quantisation is $\hat{x} = -\hbar\partial_y$.

Last but not least, we might check if $\Psi(x)$ and $\Psi^{\vee}(y)$ are related via Fourier/Laplace transformation. Using the previously mentioned identities, we compute

$$\begin{aligned}
\int \Psi(x)e^{-\frac{xy}{\hbar}}dx &= \int \varphi_{-\hbar}(x)e^{\frac{x}{2}+\frac{x\pi i}{\hbar}}e^{-\frac{xy}{\hbar}}dx \\
&= \int \tilde{\varphi}_{\sqrt{-\hbar/(2\pi i)}}(x/\sqrt{\hbar 2\pi i})e^{\frac{x}{\hbar}(\frac{\hbar}{2}+\pi i-y)}dx \\
&= \sqrt{\hbar 2\pi i}\int \tilde{\varphi}_{\sqrt{\frac{-\hbar}{2\pi i}}}(x)e^{\frac{x}{\hbar}\sqrt{\hbar 2\pi i}(\frac{\hbar}{2}+\pi i-y)}dx \\
&= \sqrt{\hbar 2\pi i}\frac{e^{-\frac{\pi i}{12}(3-\frac{\hbar}{2\pi i}-\frac{2\pi i}{\hbar})-\pi\frac{\frac{\hbar}{2}+\pi i-y}{\sqrt{\hbar 2\pi i}}(\sqrt{\frac{-\hbar}{2\pi i}}+\sqrt{\frac{2\pi i}{-\hbar}})}}{\tilde{\varphi}_{\sqrt{\frac{-\hbar}{2\pi i}}}\left(\frac{\frac{\hbar}{2}+\pi i-y}{\sqrt{\hbar 2\pi i}}+\frac{i}{2}\left(\sqrt{\frac{-\hbar}{2\pi i}}+\sqrt{\frac{2\pi i}{-\hbar}}\right)\right)} \\
&= C \cdot \frac{e^{\frac{y}{2}+\frac{y\pi i}{\hbar}}}{\varphi_{-\hbar}(-y)} = C \cdot e^{\frac{y}{2}+\frac{y\pi i}{\hbar}}\varphi_{\hbar}(y) = C \cdot \Psi^{\vee}(y),
\end{aligned}$$

where $C$ is a constant independent of $y$ and not important for us, since it would be related to the integration constant of $\Phi^{\vee}(y)$ which we have not fixed. In the first step we have inserted $\Psi(x) = \varphi_{-\hbar}(x)e^{\frac{x}{2}+\frac{x\pi i}{\hbar}}$, in the second step we have replaced $\varphi$ through $\tilde{\varphi}$ via (38), in the third step we did the variable transformation $x \to x\sqrt{\hbar 2\pi i}$, in the fourth step we applied the Fourier transform (37) of the quantum dilogarithm, in the fifth step we have used the identity $\varphi_{\hbar}(x) = \frac{1}{\varphi_{-\hbar}(x)}$ which gave finally in the last step the wave function $\Psi^{\vee}(y)$ up to an normalisation constant depending on $\hbar$.

## 3.4 Resurgence for curves of the form $e^x = F(e^y)$

Computing the wave function $\Psi(x)$ or $\Psi^{\vee}(y)$ (or more precisely the log of the wave function) from the correlator $\omega_{g,n}$ or $\omega_{g,n}^{\vee}$ give an asymptotic expansion in $\hbar$ which is in general not

convergent. The spectral curves considered in this paper of the form $e^x = F(e^y)$ possess a particular form, for which the Borel transform of the log of the wave function $\Psi^\vee(y)$ can be computed. We will essentially just apply the result [61, Prop. 2.2]. The proposition states that for a given asymptotitic series of the form

$$\phi_f(\tau, y) = \sum_{g=1}^{\infty} \frac{B_{2g}(1/2)}{(2g)!} f^{(2g)}(y)(2\pi i \tau)^{2g-1},$$

with $f$ analytic on $y$ for $|\text{Im}(y)| < \pi$, the Borel transfroam takes the form

$$\mathcal{B}[\phi_f(\tau, y)](\xi) = \frac{i}{2\pi} \sum_{g=1}^{\infty} \frac{(-1)^g}{g^2} \left( f''\left(y + \frac{\xi}{n}\right) + f''\left(y - \frac{\xi}{n}\right) \right).$$

Applying this to the previous construction of the wave function $\Psi^\vee(y)$, we set $\hbar = 2\pi i \tau$ and $f(y) = \Phi_{0,1}^\vee(y) = \int \log F(e^y) dy$ and derive the Borel transform

$$\mathcal{B}[\phi_{\Phi_{0,1}^\vee}(\frac{\hbar}{2\pi i}, y)](\frac{\xi}{2\pi i}) = \frac{i}{2\pi} \sum_{g=1}^{\infty} \frac{(-1)^g}{g^2} \left( \frac{F'(e^{y+\frac{\xi}{2\pi i n}})e^{y+\frac{\xi}{2\pi i n}}}{F(e^{y+\frac{\xi}{2\pi i n}})} + \frac{F'(e^{y-\frac{\xi}{2\pi i n}})e^{y-\frac{\xi}{2\pi i n}}}{F(e^{y-\frac{\xi}{2\pi i n}})} \right).$$

We conclude that the wave function $\Psi^\vee(y)$ takes the form (by the Laplace transform of the Borel transform, which is *not* an aymptotic expression in $\hbar$ any more)

$$\log \Psi^\vee(y) = \frac{y}{2} + \frac{i\pi}{2} + \frac{\Phi_{0,1}^\vee(y)}{\hbar} + \frac{1}{2\pi i} \int_0^\infty d\xi e^{-\xi/\hbar} \mathcal{B}\left[ \phi_{\Phi_{0,1}^\vee}\left(\frac{\hbar}{2\pi i}, y\right) \right]\left(\frac{\xi}{2\pi i}\right).$$

If we want to analyse the precise analytic properties, for instance poles in the Borel plane, the function $F$ of the spectral curve $e^x = F(e^y)$ should be specified.

## 4 Examples and consequences

### 4.1 Lambert curve

The Lambert curve is a simple example which shows already how the $x - y$ should be adjusted according to Sec. 3.2. The Lambert curve is an important example since it enumerates simple Hurwitz numbers [4] or equivalently by the ELSV formula simple Hodge integrals over $\overline{\mathcal{M}}_{g,n}$. It was observed and discussed in [26] that the spectral curve of the form

$$x(z) = z - \log z, \qquad \tilde{y}(z) = \log z \quad \rightarrow \quad x = e^{\tilde{y}} - \tilde{y}, \tag{48}$$

can be applied to the ordinary $x - y$ duality formula and generates the same $\tilde{W}_{g,n}$ as TR, see also Example 2.10 for this.

However, this curve differs from the curve of Bouchard and Marino by the symplectic transformation $\tilde{y} \rightarrow y = \tilde{y} + x$ which leaves the correlators $\tilde{W}_{g,n}$ (except for $\tilde{W}_{0,1}$) invariant (see Sec. 2.1). Note that for the curve (48) the logarithmic singularity of $x$ at $z = 0, \infty$ is cancelled by the logarithmic singularity of $\tilde{y}$ at $z = 0, \infty$ in the $x$-$y$ duality formula. The differential $\frac{d\tilde{y}}{dx}$ or $\frac{dx}{d\tilde{y}}$ in the $x - y$ formula is regular at $z = 0$ (this is not the case for $x(z) = z - \log z$ and $y(z) = z$). This cancellation has the consequence that the $x - y$ formula does not generate additional poles for $\omega_{g,n}$ or $\omega_{g,n}^\vee$ at $z = 0$. Therefore, the construction of Sec. 3.2 is not needed, and the ordinary $x - y$ formula of [17, 23] can be applied, see also Example 2.10.

Now, we change the curve via symplectic transformation $\tilde{y} \to y = \tilde{y} + x$ to

$$x(z) = z - \log z, \qquad y(z) = z \quad \to \quad e^x = \frac{e^y}{y}, \tag{49}$$

as it was originally formulated. The construction of Sec. 3.2 applies for this curve. This means that the $W_{g,1}^\vee$'s have to be corrected as suggested in Sec. 3.2 by equation (33), that is

$$
\begin{aligned}
W_{g,1}^\vee(y) &= \frac{B_{2g}(1/2)}{(2g)!} \partial_y^{2g}(y - \log y) \\
&= \delta_{g,0} y - \frac{B_{2g}(1/2)}{2g(2g-1)y^{2g}} \\
&= [\hbar^{2g-1}]\left[ \frac{y}{\hbar} - \partial_y \log \Gamma\left( \frac{y}{\hbar} + \frac{1}{2} \right) - \frac{1}{\hbar} \log \hbar + \frac{1}{2} \log 2\pi \right],
\end{aligned}
$$

where the $\Gamma$ function is understood as an asymptotic series, see Sec. 3.1. This implies that the wave function and the quantum spectral curve is of the form

$$\Psi^\vee(y) = \frac{e^{\frac{y^2}{2\hbar}} \sqrt{2\pi}}{\Gamma\left( \frac{y}{\hbar} + \frac{1}{2} \right) \hbar^{y/\hbar}},$$

$$\hat{P}^\vee(\hbar \partial_x, y) = e^{\hbar \partial_y} - \frac{e^{y + \hbar/2}}{y + \frac{\hbar}{2}}.$$

The new ingredients for the $x - y$ formula (18) considering the construction of Sec. 3.2 are

$$\hat{O}^\vee(y(z)) = \sum_{m \geq 0} \left( -\frac{\partial}{\partial x(z)} \right)^m \left( -\frac{dy(z)}{dx(z)} \right) [u^m] \frac{z^u \Gamma\left( \frac{z}{\hbar} + \frac{1-u}{2} \right)}{\hbar u \hbar^u \Gamma\left( \frac{z}{\hbar} + \frac{1+u}{2} \right)}, \tag{50}$$

$$e^{\frac{1}{2} \hat{c}^\vee(u,u,y,y)} = 1, \tag{51}$$

$$e^{\hat{c}^\vee(u_1,u_2,y(z_1),y(z_2))} = \frac{(z_1 - \hbar u_1/2 - z_2 + \hbar u_2/2)(z_1 + \hbar u_1/2 - z_2 - \hbar u_2/2)}{(z_1 + \hbar u_1/2 - z_2 + \hbar u_2/2)(z_1 - \hbar u_1/2 - z_2 - \hbar u_2/2)}. \tag{52}$$

Showing that either (30) of Example 2.10, or (50), (51) and (52) give the same $\tilde{W}_{g,n} = W_{g,n}$ after inserting in (18) is a non trivial task, but follows after long computations which were performed in [17] in a slightly more general setting. We explicitly write down

$$
\begin{aligned}
W_{g,n}(x_1(z_1), ..., x_n(z_n)) &= \prod_{i=1}^n \left( \sum_{m_i \geq 0} \left( -\frac{\partial}{\partial x_i(z_i)} \right)^{m_i} \left( -\frac{dy_i(z_i)}{dx_i(z_i)} \right) [u_i^{m_i}] \frac{z_i^{u_i} \Gamma\left( \frac{z_i}{\hbar} + \frac{1-u_i}{2} \right)}{\hbar^{u_i} \Gamma\left( \frac{z_i}{\hbar} + \frac{1+u_i}{2} \right)} \right) \\
&\quad \times \sum_{\substack{\sigma \in S_n \\ \sigma = n\text{-cycle}}} \prod_{i=1}^n \frac{1}{z_i + \frac{\hbar u_i}{2} - z_{\sigma(i)} + \frac{\hbar u_{\sigma(i)}}{2}},
\end{aligned}
$$

where we have used that the sum over all graphs $\mathcal{G}_n^2$ can be represented as sum over $n$-cycles (see Proposition 2.7).

Taking the Laplace transform of this equation (as explained in [26]), the following formula for the linear Hodge integrals is deduced

$$\prod_{i=1}^{n} \frac{k_i^{k_i+1}}{k_i!} \left\langle \frac{\Lambda(1)}{\prod_{i=1}^{n}(1-k_i\psi_i)} \right\rangle_{g,n} = \operatorname{Res}_{z_i=0}[\hbar^{2g+n-2}] \prod_{i=1}^{n} \frac{dz_i e^{k_i z_i}}{\hbar^{k_i}} \frac{\Gamma\left(\frac{z_i}{\hbar} + \frac{1-k_i}{2}\right)}{\Gamma\left(\frac{z_i}{\hbar} + \frac{1+k_i}{2}\right)}$$

$$\times \sum_{\substack{\sigma \in S_n \\ \sigma = n\text{-cycle}}} \prod_{i=1}^{n} \frac{1}{z_i + \frac{\hbar k_i}{2} - z_{\sigma(i)} + \frac{\hbar k_{\sigma(i)}}{2}},$$

where for $n=1$ we define $\sum_{\substack{\sigma \in S_1 \\ \sigma = 1\text{-cycle}}} \frac{1}{z_i + \frac{\hbar k_i}{2} - z_{\sigma(i)} + \frac{\hbar k_{\sigma(i)}}{2}} = \frac{1}{\hbar u_i}$.

**Remark 4.1.** *A generalisation of these linear Hodge integrals to integrals of the form* $\left\langle \frac{\Lambda(a)}{\prod_{i=1}^{n}(1-k_i\psi_i)} \right\rangle_{g,n}$ *was for instance considered in [63] and proved to satisfy TR with the curve* $e^{ax} = \frac{y}{e^{ay}}$ *in [6, 64]. From the Hurwitz point of view, this count of ramified coverings is called orbifold Hurwitz numbers (the ramification profile over infinity is* $\mu = (\mu_1, ..., \mu_n)$ *and over 0 of the form* $(a, ...., a)$, *see [6] for details). The corresponding spectral curve fits perfectly in the class of curves under consideration. It is straightforward to derive a formula for* $\left\langle \frac{\Lambda(a)}{\prod_{i=1}^{n}(1-k_i\psi_i)} \right\rangle_{g,n}$ *from the $x - y$ duality formula for any $a$.*

## 4.2 The framed topological vertex curve

The framed Topological Vertex curve encodes the Gromov-Witten invariants of $\mathbb{C}^3$, more precisely the framed mirror curve of $\mathbb{C}^3$ is the curve

$$e^x = \frac{e^{yf}}{(1-e^{-y})},$$

with framing $f$. This curve is important in topological string theory. It was conjectured in [11] more generally that if one takes the mirror curve of a toric Calabi-Yau 3-fold to be the spectral curve, the $\tilde{\omega}_{g,n}$ generated by TR computes the $B$-model correlators. In the specific case of $\mathbb{C}^3$, the conjecture was proved for instance in [12, 65] and in general in [13].

However note that for the special framing $f = 0$, TR does not give the expected results neither for the $\tilde{\omega}_{g,n}$ nor for the free energies, which was observed for instance in [40].

We take the following parametrisation of the curve

$$x(z) = -f \log z - \log(1-z), \qquad y(z) = -\log z.$$

The correlators computed by TR (1) from this curve yield triple Hodge integrals on $\overline{\mathcal{M}}_{g,n}$ [65, Theorem 4.1]

$$\tilde{\omega}_{g,n}(z_1, ..., z_n) = (f(1+f))^{g-1} \sum_{\mu_1, ..., \mu_n} \prod_{i=1}^{n} \frac{(\mu_i(1+f))!}{\mu_i!(f\mu_i)!} e^{-\mu_i x_i(z_i)} \mu_i dx_i(z_i) \left\langle \frac{\Lambda(1)\Lambda(f)\Lambda(-1-f)}{\prod_{i=1}^{n}(1+\mu_i\psi_i)} \right\rangle_{g,n},$$

where $\psi_i$ is the $i$th $\psi$-class and $\Lambda(\alpha) = 1 + \sum_k (-1)^k \alpha^{-k} c_k(\mathbb{E})$ with $c_k$ the $k$th Chern class of the Hodge bundle $\mathbb{E}$.

Taking the Laplace transform of this with contour in the $z$-plane around the origin yields (since $\operatorname{Res}_{z_i=1} e^{x_i(z_i)(k_i-\mu_i)} dx_i(z_i) = -\delta_{\mu_i, k_i} f$)

$$\operatorname{Res}_{z_i=0} e^{x_i(z_i)k_i} \tilde{\omega}_{g,n}(z_1, ..., z_n) = (f(1+f))^{g-1} \prod_{i=1}^{n} \left(-\frac{(k_i(1+f))! f k_i}{k_i!(f k_i)!}\right) \left\langle \frac{\Lambda(1)\Lambda(f)\Lambda(-1-f)}{\prod_{i=1}^{n}(1+k_i\psi_i)} \right\rangle_{g,n}. \quad (53)$$

Now, we want to apply the construction of Sec. 3.2 which is also valid for the case of $f = 0, -1$. We find

$$W_{0,1}^\vee(y) = f y - \log(1 - e^{-y}), \qquad W_{g,1}^\vee(y) = -\frac{B_{2g}(1/2)}{(2g)!} \partial_y^{2g} \log(1 - e^{-y}). \tag{54}$$

By construction and the argumentation of Sec. 3.2, the wave function $\Psi^\vee(y)$ and the quantum spectral curve $\hat{P}^\vee$ are as expected from eq. (13).

The ingredients for the $x - y$ formula (18) considering (54) for $W_{g,n}^\vee$ are

$$\hat{O}^\vee(y(z)) = \sum_{m \geq 0} \left(-\frac{\partial}{\partial x(z)}\right)^m \left(-\frac{dy(z)}{dx(z)}\right) [u^m] \frac{\exp\left[\left(\frac{S(\hbar u \partial_{y(z)})}{S(\hbar \partial_{y(z)})} - 1\right) u x(z)\right]}{\hbar u}, \tag{55}$$

$$e^{\frac{1}{2}\hat{c}^\vee(u,u,y,y)} = \frac{1}{S(\hbar u)}, \tag{56}$$

$$e^{\hat{c}^\vee(u_1,u_2,y(z_1),y(z_2))} = \frac{\left(z_1 e^{-\hbar u_1/2} - z_2 e^{-\hbar u_2/2}\right)\left(z_1 e^{\hbar u_1/2} - z_2 e^{\hbar u_2/2}\right)}{\left(z_1 e^{\hbar u_1/2} - z_2 e^{-\hbar u_2/2}\right)\left(z_1 e^{-\hbar u_1/2} - z_2 e^{\hbar u_2/2}\right)}. \tag{57}$$

Showing that (55), (56) and (57) give the same $\tilde{W}_{g,n} = W_{g,n}$ after inserting in (18) is again a non trivial task, but tested with computer algebra system for several examples. We explicitly write down the formula following from the $x - y$ transformation formula

$$W_{g,n}(x_1(z_1), ..., x_n(z_n))$$
$$= \prod_{i=1}^n \left\{ \sum_{m_i \geq 0} \left(-\frac{\partial}{\partial x_i(z_i)}\right)^{m_i} \left(-\frac{dy_i(z_i)}{dx_i(z_i)}\right) [u_i^{m_i}]_{z_i} \exp\left[\left(\frac{S(\hbar u_i \partial_{y_i(z_i)})}{S(\hbar \partial_{y_i(z_i)})} - 1\right) u_i x_i(z_i)\right] \right\}$$
$$\times \sum_{\substack{\sigma \in S_n \\ \sigma = n\text{-cycle}}} \prod_{i=1}^n \frac{1}{z_i e^{\hbar u_i/2} - z_{\sigma(i)} e^{-\hbar u_{\sigma(i)}/2}},$$

where we have used that the sum over all graphs $\mathcal{G}_n^2$ can be represented as sum over $n$-cycles (see Proposition 2.7) and the Cauchy matrix in the proof Proposition 2.7 has to be chosen with $a_i = z_i e^{\hbar u_i/2}$ and $b_j = -z_j e^{-\hbar u_j/2}$.

Taking the Laplace transform of this equation (as explained in [26]), the following formula for the triple Hodge integrals is deduced

$$(f(1+f))^{g-1} \prod_{i=1}^n \left(-\frac{(k_i(1+f))! f k_i}{k_i!(f k_i)!}\right) \left\langle \frac{\Lambda(1)\Lambda(f)\Lambda(-1-f)}{\prod_{i=1}^n (1 + k_i \psi_i)} \right\rangle_{g,n}$$
$$= \text{Res}_{z_i=0}[\hbar^{2g+n-2}] \prod_{i=1}^n dz_i \, e^{\frac{S(\hbar k_i \partial_{y_i(z_i)})}{S(\hbar \partial_{y_i(z_i)})} k_i x_i(z_i)} \sum_{\substack{\sigma \in S_n \\ \sigma = n\text{-cycle}}} \prod_{i=1}^n \frac{1}{z_i e^{\hbar k_i/2} - z_{\sigma(i)} e^{-\hbar k_{\sigma(i)}/2}},$$

where we remind $S(t) = \frac{e^{t/2} - e^{-t/2}}{t}$, $x(z) = -f \log z - \log(1 - z)$ and $y(z) = -\log z$, and $f$ is the framing. This formula seems to be of a very similar nature as the original Marino-Vafa formula [66]. In the case $n = 1$, we have to take $\sum_{\substack{\sigma \in S_1 \\ \sigma = 1\text{-cycle}}} \frac{1}{z_i e^{\hbar k_i/2} - z_{\sigma(i)} e^{-\hbar k_{\sigma(i)}/2}} = \frac{1}{z_i \hbar u_i S(\hbar u_i)}$.

One might further simplify the expression in the following way

$$\exp\left(\frac{S(\hbar k_i \partial_{y_i(z_i)})}{S(\hbar \partial_{y_i(z_i)})} k_i x_i(z_i)\right) = \exp\left(\sum_{n=0}^{k_i-1} e^{\hbar(n + \frac{1-k_i}{2}) z_i \partial_{z_i}}(-f \log z_i - \log 1 - z_i)\right)$$
$$= \frac{1}{z_i^{f k_i} \prod_{n=0}^{k_i-1}(1 - z_i e^{\hbar(n + \frac{1-k_i}{2})})},$$

where the formal action $a^{z\partial_z} f(z) = f(az)$ was used. For instance for $n = 1$, we find

$$(f(1+f))^{g-1}\left(-\frac{(k(1+f))!\,f\,k}{k!(f\,k)!}\right)\left\langle\frac{\Lambda(1)\Lambda(f)\Lambda(-1-f)}{(1+k\psi)}\right\rangle_{g,1}$$
$$= \operatorname{Res}_{z=0}[\hbar^{2g-1}]\frac{dz}{(e^{\hbar k/2} - e^{-\hbar k/2})z^{f\,k+1}\prod_{n=0}^{k-1}(1 - ze^{\hbar(n+\frac{1-k}{2})})}.$$

**Remark 4.2.** *For more general triple Hodge integrals of Calabi-Yau condition of the form*

$$\left\langle\frac{\Lambda(p)\Lambda(q)\Lambda(-p-q)}{\prod_{i=1}^n(1+k_i\psi_i)}\right\rangle_{g,n}, \tag{58}$$

*it is also known that they can be computed by TR, see [67,68]. The spectral curve for these integrals is also of the form $e^x = F(e^y)$ such that a formula for the more general triple Hodge integrals of the form (58) can also be written down. However, formulas are getting pretty ugly and there is no new insight for $\Lambda(p)\Lambda(q)\Lambda(-p-q)$ since $\Lambda(1)\Lambda(f)\Lambda(-1-f)$ has the same enumerative geometric content.*

**Remark 4.3.** *Adding additional $\Theta$-classes as they are introduced by Norbury [45] to triple Hodge integrals can be performed as well, where the spectral curve is known [68] and again of the form $e^x = F(e^y)$. These are called triple $\Theta$-Hodge integrals. More simple intersections numbers where mixtures of a $\Theta$-class just with $\Psi$-classes have a spectral curve of the form $P(x,y) = xy^r - 1 = 0$ [46]. Results of Sec. 2.5 from the $x-y$ duality with unramified $y$ are applicable here, a particular example is the Bessel curve of the form $(\mathbb{P}^1, z^2, \frac{1}{z}, \frac{dz_1\,dz_2}{(z_1-z_2)^2})$ [45].*

## 4.3 Descendent Gromov-Witten invariants of $\mathbb{P}^1$

The stationary Gromov-Witten invariants on $\mathbb{P}^1$ with their relation to integrable systems were considered by Pandharipande in [69] and Okounkov and Pandharipande in [70]. Later, it was conjectured that these invariants are computable by TR in [71], which was proved in [72]. The stationary Gromov-Witten invariants are defined by (see for instance [70] and references therein)

$$\left\langle\prod_{i=1}^n\tau_{b_i}(\gamma)\right\rangle_g^d = \int_{[\overline{\mathcal{M}}_{g,n}(\mathbb{P}^1,d)]^{vir}}\prod_{i=1}^n\psi_i^{b_i}ev_i^*(\gamma), \tag{59}$$

where $d$ satisfies the dimension condition $\sum_i b_i = 2g - 2 + 2d$, $\gamma \in H^2(\mathbb{P}^1)$ be the dual class of a point and $\overline{\mathcal{M}}_{g,n}(\mathbb{P}^1,d)$ is the moduli space of degree $d$ maps from a Riemann surface of genus $g$ with $n$ marked points to $\mathbb{P}^1$. The corresponding spectral curve computing stationary Gromov-Witten invariants is

$$x = e^y + e^{-y}, \tag{60}$$

which is typically parametrised by

$$x(z) = z + \frac{1}{z}, \qquad y(z) = \log z.$$

The precise statement in the notation of the article, where $\tilde{\omega}_{g,n}(z_1,...,z_n)$ is derived by TR, is the following [72]:

$$\tilde{\omega}_{g,n}(z_1,...,z_n) = \sum_{b_1,...,b_n}\left\langle\prod_{i=1}^n\tau_{b_i}(\gamma)\right\rangle_g^d\prod_{i=1}^n\frac{(b_1+1)!}{x(z_i)^{b_i+2}}dx(z_i).$$

The equality holds as a Laurent expansion at $x_i \to \infty$.

We will now discuss that taking this curve and the (conjectured) application of the $x - y$ formula together with the proposition of this article is compatible, also with taking singular limits, for instance, of colliding ramification points. This will further support the proposed structure of Sec. 3.2. The curve (60) does not fit into the class of curves considered in Sec. 3.2 (also not after interchanging $x$ and $y$). However if we add a parameter $t$ by

$$x_t(z) = z + \frac{t^2}{z}, \qquad y(z) = \log z \quad \to \quad x_t = e^y + t^2 e^{-y}, \tag{61}$$

we get a new curve which coincides with the spectral curve (31) of Sec. 3.1 for $t \to 0$ and with the curve (60) for $t = 1$.

The limit $t \to 0$ is a singular limit, since the two ramification points at $\beta_\pm = \pm t$ merge and cancel out, and on the other hand the ramification points merge with the logarithmic singularity of $y(z)$. This geometry of the limit is not included in the recent work [73] where several different cases of a $t \to 0$ limit are discussed.

However, taking the $x - y$ duality formula into account and the construction of Sec. 3.2 the limit $t \to 0$ from the curve (61) and from the construction of Sec. 3.2 (after changing the role of $x$ and $y$) coincide. To see this, we observe first that the curve (61) does not fit into the class considered in Sec. 3.2 for $t \neq 0$. However, in both cases $t = 0$ and $t \neq 0$ we have $\tilde{\omega}^\vee_{g,n} = 0$ for $2g + n - 2 > 0$ since $y(z)$ has no ramification points. Following the $x - y$ formula (18), we have the ingredients

$$\hat{O}^\vee(y(z)) = \sum_{m \geq 0} \left( -\frac{\partial}{\partial x_t(z)} \right)^m \left( -\frac{dy(z)}{dx_t(z)} \right) [u^m] \frac{\exp\left[ u \left( S(\hbar u) - 1 \right) \left( z + \frac{t^2}{z} \right) \right]}{\hbar u}, \tag{62}$$

$$e^{\frac{1}{2} \hat{c}^\vee(u,u,y,y)} = \frac{1}{S(\hbar u)}, \tag{63}$$

$$e^{\hat{c}^\vee(u_1,u_2,y(z_1),y(z_2))} = \frac{\left( z_1 e^{-\hbar u_1/2} - z_2 e^{-\hbar u_2/2} \right) \left( z_1 e^{\hbar u_1/2} - z_2 e^{\hbar u_2/2} \right)}{\left( z_1 e^{\hbar u_1/2} - z_2 e^{-\hbar u_2/2} \right) \left( z_1 e^{-\hbar u_1/2} - z_2 e^{\hbar u_2/2} \right)}. \tag{64}$$

We conclude from the $x - y$ duality formula (18) (or more precisely a version of the special case (19) where $y = \log z$ which just conjecturally holds[6]) together with ideas of the proof of Proposition 2.7 (as explained in Example 2.10)

$$\tilde{W}_{g,n}(x_t(z_1), ..., x_t(z_n))$$
$$= \prod_{i=1}^{n} \left\{ \sum_{m_i \geq 0} \left( -\frac{\partial}{\partial x_t(z_i)} \right)^{m_i} \left( -\frac{dy(z_i)}{dx_t(z_i)} \right) [u_i^{m_i}] z_i \exp\left[ u_i \left( S(\hbar u_i) - 1 \right) \left( z_i + \frac{t^2}{z_i} \right) \right] \right\}$$
$$\times \sum_{\substack{\sigma \in S_n \\ \sigma = n\text{-cycle}}} \prod_{i=1}^{n} \frac{1}{z_i e^{\hbar u_i/2} - z_{\sigma(i)} e^{-\hbar u_{\sigma(i)}/2}}, \tag{65}$$

which was tested with computer algebra systems for small $g$ and $n$. Note that the rhs behaves smoothly in $t$, which converges for $t \to 0$ to the case considered in Sec. 3.1. However, the derivation of $W_{g,n}(x_t(z_1), ..., x_t(z_n))$ from TR is not smooth in $t$ since ramification points collide and merge with the singularity of the logarithm. This underpins again that TR needs some adaption if the curve is of the form $e^y = F(e^x)$ or $e^y = F(x)e^{ax}$, respectively.

---

[6]Later proved in [29].

Since we can extract from the correlators $\tilde{\omega}_{g,n}$ the stationary Gromov-Witten invariants via Laplace transform

$$(-1)^n \operatorname{Res}_{z_i=0} dx(z_1)...dx(z_n) e^{\mu_1 x(z_1)+...+\mu_n x(z_n)} \tilde{\omega}_{g,n} = \sum_{b_i} \left\langle \prod_{i=1}^n \tau_{b_i}(\gamma) \right\rangle_g^d \prod_i \mu_i^{b_i+1}$$

$$= \int_{[\overline{\mathcal{M}}_{g,n}(\mathbb{P}^1,d)]^{vir}} ev_i^*(\gamma) \prod_{i=1}^n \frac{\mu_i}{1-\mu_i\psi_i},$$

we derive the following formula for the stationary Gromov-Witten invariants (59) from the Laplace transform of the $x-y$ duality formula (65) (holds just conjecturally) with $t=1$

$$\int_{[\overline{\mathcal{M}}_{g,n}(\mathbb{P}^1,d)]^{vir}} ev_i^*(\gamma) \prod_{i=1}^n \frac{\mu_i}{1-\mu_i\psi_i} = [\hbar^{2g+n-2}] \prod_{i=1}^n \operatorname{Res}_{z_i=0} dz_i e^{\mu_i S(\hbar\mu_i)(z_i+\frac{1}{z_i})}$$

$$\times \sum_{\substack{\sigma \in S_n \\ \sigma=n\text{-cycle}}} \prod_{i=1}^n \frac{1}{z_i e^{\hbar\mu_i/2} - z_{\sigma(i)} e^{-\hbar\mu_{\sigma(i)}/2}},$$

and the order of integration is the same for each summand of the sum over $\sigma \in S_n$.

**Example 4.4.** *Let us consider the $n=1$ example of the above formula:*

$$\int_{[\overline{\mathcal{M}}_{g,n}(\mathbb{P}^1,d)]^{vir}} ev_i^*(\gamma) \frac{\mu}{1-\mu\psi} = [\hbar^{2g-1}] \operatorname{Res}_{z=0} \frac{dz e^{\mu S(\hbar\mu)(z+\frac{1}{z})}}{z\hbar\mu S(\hbar\mu)}$$

$$= [\hbar^{2g-1}] \operatorname{Res}_{z=0} dz \sum_{n=0}^\infty \frac{(\mu S(\hbar\mu)(z+\frac{1}{z}))^n}{n! z\hbar\mu S(\hbar\mu)}$$

$$= [\hbar^{2g}] \sum_{n=0}^\infty \frac{\mu^{n-1} S(\hbar\mu)^{n-1}}{(\frac{n}{2})!(\frac{n}{2})!}.$$

*Taking on the lhs due to dimensional restriction the coefficient $\mu^{2g+2d-1}$ yields the formula of Pandharipande [69, Theorem 1] with $n=2d$.*

**Example 4.5.** *The degree $d=1$ case is also an important explicit special case (considered by Pandharipande in [69, Theorem 2]) which can easily be derived from the above formula. Due to dimensional restriction, we have after expanding the geometric series $\frac{1}{1-\mu_i\psi_i} = \sum_{b_i} \mu_i^{b_i}\psi_i^{b_i}$ that $\sum_i b_i = 2g$. Taking the $[\mu_1^{b_1+1}...\mu_n^{b_n+1}]$ coefficient, we have on the lhs*

$$\int_{[\overline{\mathcal{M}}_{g,n}(\mathbb{P}^1,1)]^{vir}} ev_i^*(\gamma) \prod_i \psi_i^{b_i}.$$

*Since we have to take on the rhs the coefficient $[\hbar^{2g+n-2}]$ and $[\mu_1^{b_1+1}...\mu_n^{b_n+1}]$ with $\sum_i(b_i+1) = 2g+n$, we can redefine all $\mu_i \to \frac{\mu_i}{\hbar}$ on the rhs. The remaining coefficient in $\hbar$ is $[\hbar^{-2}]$, which can just be generated by expanding at most two exponentials*

$$[\hbar^{-2}][\mu_1^{b_1+1}...\mu_n^{b_n+1}] \prod_{i=1}^n \operatorname{Res}_{z_i=0} dz_i e^{\frac{\mu_i}{\hbar} S(\mu_i)(z_i+\frac{1}{z_i})} \sum_{\substack{\sigma \in S_n \\ \sigma=n\text{-cycle}}} \prod_{i=1}^n \frac{1}{z_i e^{\mu_i/2} - z_{\sigma(i)} e^{-\mu_{\sigma(i)}/2}}.$$

*Now, it is important that the order of the contour integrals is fixed, since swapping order of the contour integrals gives nontrivial contributions of the form*

$\mathrm{Res}_{z_i=0}\,\mathrm{Res}_{z_j=0}=\mathrm{Res}_{z_j=0}\,\mathrm{Res}_{z_i=0}+\mathrm{Res}_{z_j=0}\,\mathrm{Res}_{z_i=z_j}$. *Therefore, we fix for all permutations $\sigma\in S_n$ the order of the integrals in the consecutive order, i.e. first $z_1$, then $z_2$ etc. It is easy to see that the residue for $z_1$ already vanishes except if the exponential $e^{\frac{\mu_1}{\hbar}S(\mu_1)(z_1+\frac{1}{z_1})}$ is expanded. Furthermore, since the last contour integral is $z_n$, we have to expand also $e^{\frac{\mu_n}{\hbar}S(\mu_n)(z_n+\frac{1}{z_n})}$ otherwise the expression vanishes. This means that fixing the order of integration and taking the $[\hbar^{-2}]$ coefficient yields just nontrivial contribution of the form*

$$[\mu_1^{b_1+1}...\mu_n^{b_n+1}]\,\mathrm{Res}_{z_n=0}\,dz_n...\mathrm{Res}_{z_1=0}\,dz_1\mu_1 S(\mu_1)\left(z_1+\frac{1}{z_1}\right)\mu_n S(\mu_n)\left(z_n+\frac{1}{z_n}\right)$$

$$\times\sum_{\substack{\sigma\in S_n\\\sigma=n\text{-cycle}}}\prod_{i=1}^{n}\frac{1}{z_i e^{\mu_i/2}-z_{\sigma(i)}e^{-\mu_{\sigma(i)}/2}}\,.$$

*Now, computing the residues for all $(n-1)!$ permutations $\sigma\in S_n$ with $\sigma$ an n-cycle, a lot of permutations have vanishing contributions except for $2^{n-2}$ permutations $\sigma\in S_n$. This $2^{n-2}$ permutations can be characterised nicely in the cycle notation as follows. Put 1 on the first place, than we have to put 2 either at the first or last empty space*

$$(1,2,....)\,,\qquad or\qquad (1,....,2)\,,$$

*otherwise the integral vanishes. Adding 3 again either at the first or last empty space:*

$$(1,2,3....)\,,\qquad or\qquad (1,2,....,3)\,,\qquad or\qquad (1,3,....,2)\,.\qquad or\qquad (1,....,3,2)\,,$$

*etc. Constructing the subset of $2^{n-2}$ permutations $\sigma\in S_n$ in this way, these are the only permutations which do not vanish if we take the order of integration to be the consecutive order. Performing the integration, each permutation gives a term of the form $(\pm)^{\epsilon_2+...+\epsilon_{n-1}}e^{\epsilon_2\frac{\hbar\mu_2}{2}+\epsilon_3\frac{\hbar\mu_3}{2}...+\epsilon_{n-1}\frac{\hbar\mu_{n-1}}{2}}$, where $\epsilon_i=\pm$ depends on if we have put i at the first or last empty space by constructing the permutation. Therefore, we find*

$$\int_{[\overline{\mathcal{M}}_{g,n}(\mathbb{P}^1,1)]^{vir}}ev_i^*(\gamma)\prod_i\psi_i^{b_i}$$

$$=[\mu_1^{b_1+1}...\mu_n^{b_n+1}]\,\mathrm{Res}_{z_n=0}\,dz_n...\mathrm{Res}_{z_1=0}\,dz_1\mu_1 S(\mu_1)\left(z_1+\frac{1}{z_1}\right)\mu_n S(\mu_n)\left(z_n+\frac{1}{z_n}\right)$$

$$\times\sum_{\substack{\sigma\in S_n\\\sigma=n\text{-cycle}}}\prod_{i=1}^{n}\frac{1}{z_i e^{\mu_i/2}-z_{\sigma(i)}e^{-\mu_{\sigma(i)}/2}}$$

$$=[\mu_1^{b_1+1}...\mu_n^{b_n+1}]\mu_1 S(\mu_1)\mu_n S(\mu_n)\sum_{(\epsilon_2,...,\epsilon_{n-1})\in\{+1,-1\}^{n-1}}\prod_{i=2}^{n-1}(-1)^{\epsilon_i}e^{\epsilon_i\hbar\mu_i/2}$$

$$=[\mu_1^{b_1+1}...\mu_n^{b_n+1}]\prod_{i=1}^{n}(e^{\hbar\mu_i/2}-e^{-\hbar\mu_i/2})$$

$$=\prod_{i=1}^{n}\frac{1}{2^{2b_i}(2b_i+1)!}\,,$$

*where $\sum_i b_i=2g$. This result obviously coincides with [69, Theorem 2] as claimed before.*

## 5  Outlook

This article presents new ideas on how the universal $x-y$ duality of the theory of TR can be applied to a simple class of spectral curves of the form $e^x=F(e^y)$ (or $e^x=F(y)e^{ay}$) with exponential variables. These types of curves find applications in topological string theory, where

the spectral curve corresponds to the mirror of the toric Calabi-Yau 3-fold, and the generated correlators are the $B$-model correlators. The main idea was to define the (a priori trivial family) $\omega_{g,n}^{\vee} := \delta_{n,1} \frac{B_{2g}(1/2)}{(2g)!} \partial_y^{2g} \omega_{0,1}^{\vee}$. The same coefficients appear in the asymptotic expansion of the quantum dilogarithm, providing a primary example. Further examples discussed in this article include the framed topological vertex and stationary Gromov-Witten invariants on $\mathbb{P}^1$. Additionally, in the context of polynomial spectral curves $P(x, y) = 0$, we provide further examples of the $x-y$ duality formula in Section 2.5, relating it structurally to the determinantal formula in the case where $y$ is unramified.

From here, several further directions can be pursued:

- We have seen in Sec. 2.5 that the $x-y$ duality formula can be brought into the same form as the determinantal formula if $y$ is unramified via the Cauchy determinant. However, bringing the general $x - y$ formula into such a form, where the sum over all graphs $\mathcal{G}_n$ is turned into a sum over $n$-cycle permutations, is absolutely nontrivial but might be possible. The advantage is that the sum over graphs $\mathcal{G}_n$ includes many more terms since the number of these graphs grows much faster than the number of permutations. Furthermore, having such an $x - y$ duality formula will provide a new way to express the kernel $K(x_i, x_j)$ through the family $\tilde{\omega}_{g,n}^{\vee}$ rather than through the family $\tilde{\omega}_{g,n}$.

- The $\hbar$ series of the family $\tilde{\omega}_{g,n}$ is factorially growing. The determinantal formula can be used to compute the Borel transform. It is worth investigating whether the $x-y$ duality formula can also be used to derive the Borel transform or make some other asymptotic predictions.

- The two wave functions, $\tilde{\Psi}(x)$ and $\tilde{\Psi}^{\vee}(y)$, related by $x - y$ duality, are heuristically Fourier/Laplace transformations of each other. The $x - y$ duality between the two families, $\omega_{g,n}$ and $\omega_{g,n}^{\vee}$, should recover this Fourier/Laplace transformation at each order in a formal $\hbar$ expansion. Making this observation more rigorous will interplay with resurgence since the Fourier/Laplace transformation is in principle defined for a non-vanishing parameter $\hbar$.

- Extending the $x-y$ duality formula to general spectral curves with exponential variables of the form $P(e^x, e^y) = 0$, especially with higher genus, will significantly broaden its range of applications.

# Acknowledgments

I would like to thank Nitin Chidambaram, Alessandro Giacchetto, Reinier Kramer, Motohico Mulase and Federico Zerbini for discussions. I am especially grateful to Alexander Alexandrov, Boris Bychkov, Petr Dunin-Barkwoski, Maxim Kazarian and Sergey Shadrin on comments on my unpublished draft and explaining that the proposed construction appeared, in principle, already in some of their earlier papers [1,2], but not recognised in this way. I am also grateful to an anonymous referee for suggestions.

**Funding information** This work was supported through the Walter-Benjamin fellowship, funded by the Deutsche Forschungsgemeinschaft (DFG, German Research Foundation) – Project-ID 465029630.

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
