# Peer review of "$x-y$ duality in Topological Recursion for exponential variables via Quantum Dilogarithm"

_SciPost Physics, doi:SciPost Phys. 17, 065 (2024)_

## Round 1 · Referee Report · Anonymous (Referee 1) · 2024-3-27

Strengths
-
New innovative idea of deforming the topological recursion in a way that makes it compatible with the x-y duality formula beyond the known algebraic case.
-
Many new and known formulas reproduced as a direct application of this new idea.
-
The paper has already enriched the realm of topological recursion and inspired new research in this area.
Weaknesses
Report
Requested changes
No changes are requested.

---

## Round 1 · Referee Report · Anonymous (Referee 2) · 2024-6-17

Report
The current paper proposes a small step to further advance the study of x-y duality. In recent papers, the x-y duality formula is proved for algebraic spectral curves with simple ramification. Algebraic here means that x and y can be though of as meromorphic functions on a compact Riemann surface.
In the current paper, the author proposes a study of a certain class of spectral curves in exponential variables; in this case, x and y may not be meromorphic on a compact Riemann surface, they may have log singularities. When this happens, the x-y duality formula may fail if the usual topological recursion is applied to the spectral curve and its x-y dual. The author then asks the following question: how can topological recursion be modified for this class of curves so that the x-y duality formula remains valid?
The main result of the paper is a conjectural answer to this question. The author proposes a modification of topological recursion (adding extra contributions to some of the correlators) for this class of curve. The conjecture then is that these modified correlators satisfy the usual x-y duality formula. The claim is not proved in this paper, but it is checked in some examples, either by recovering known formulae in special cases, or using a computer algebra system.
The paper would be stronger if it was proven that the proposed modification of topological recursion does indeed imply that the x-y duality formula holds for this class of curves. Nevertheless, the conjectured proposal is a significant contribution to the study of x-y duality and deserves publication. Moreover, the examples studied are interesting in enumerative geometry, and the obtained formulae may be interesting in this context as well. As a result, I strongly suggest publication.
Requested changes
Warnings issued while processing user-supplied markup:
- Inconsistency: plain/Markdown and reStructuredText syntaxes are mixed. Markdown will be used.
Add "#coerce:reST" or "#coerce:plain" as the first line of your text to force reStructuredText or no markup.
You may also contact the helpdesk if the formatting is incorrect and you are unable to edit your text.
The paper contains many typos, grammatical mistakes and some imprecise statements, which I list below.
Abstract: the enumeration in "... for instance, the topological vertex curve which computes Gromov-Witten invariants of $\mathbb{C}^3$, equivalently triple Hodge integrals on the moduli spaace of complex curves, orbifold Hurwitz numbers, or stationay Gromov-Witten invariants of $\mathbb{P}^1$." is a bit confusing. Perhaps it would read better as something like "... for instance, Gromov-Witten invariants of $\mathbb{C}^3$ (or, equivalently, triple Hodge integrals), orbifold Hurwitz numbers, and stationary Gromov-Witten invariants of $\mathbb{P}^1$."
p.1: "... can be seen as an algorithm that associates to the a complex curve..." should be "... can be seen as an algorithm that associates to a complex curve..."
p.1: "... (vanishing locus of a polynomial in two variables $P(x,y)=0$)..." is too restrictive; in fact, most spectral curves studied in this paper are not of this form. Perhaps this statement in brackets should simply be removed, as it is too restrictive and this sentence is meant to be just a rough explantion of the topological recursion framework.
p.1: The sentence "Therefore, being able to express one family explicitly depends somehow on the other family." is a little confusing. I believe that what the author means here is something like "Therefore, it appears that the existence of simple explicit formulae for the multi-differentials $\omega_{g,n}$ depends on the properties of the dual spectral curve with x and y exchanged."
p.2: "... (equivalently to linear Hodge integrals) ..." should be "... (equivalently, linear Hodge integrals)..."
p.2: "... (equivalently to triple Hodge integrals..." should be "... (equivalently, triple Hodge integrals ..."
p.2: "This proposed redefinition of $\omega^\vee_{g,n}$ makes the $x-y$ duality formula work again, which is tested on several examples." Just to be precise, it may be better to clarify here that this is a conjecture and not proved in the current paper. Perhaps something like "It is conjectured that this proposed redefinition of $\omega^\vee_{g,n}$ makes the $x-y$ duality formula work again, which is tested on several examples."
p.3: "... for simple Hurwitz numbers or equivalently by the ELSV formula for linear Hodge integrals." should perhaps be something like "... for simple Hurwitz numbers or, equivalently, for linear Hodge integrals (using the ELSV formula)."
p.4: "... the extension to spectral curves of the for ..." should be "... the extension to spectral curves of the form ..."
p.4: "... is a not necessarily compact Riemann surface with $x,y: \Sigma \to \mathbb{C}$ are complex functions such that..." should be "... is a not necessarily compact Riemann surface, with $x,y: \Sigma \to \mathbb{C}$ complext functions such that ..."
p.4: Second paragraph: the author wants to allow log singularities for x and y. The way he defines the spectral curve is for x and y to be complex functions (not necessarily meromorphic) on the Riemann surface $\Sigma$, but such that $dx$ and $dy$ are meromorphic on $\Sigma$. But then he writes: "Both functions x,y should have at most simple ramification points on $\Sigma$ which are distinct ...". Perhaps it would be better to explicitly say something like "Both functions $x,y$ should have at most simple ramifications and log singularities ... " (I also removed which are distinct, as I am not sure what non-distinct ramification points are... wouldn't they just be a ramification point again?)
p.4: Below the statement of topological recursion, the notation should be defined a little more precisely. It should be stated that the subsets $I_1$ and $I_2$ must be disjoint but can be empty, and the strict inclusion $I_1, I_2 \subset I$ should be $I_1, I_2 \subseteq I$.
p.4: "the ramification points $\eta_i$ of $x$ are defined by $dx(\beta_i)= 0$." This is only a subset of the ramification points of x; poles of x of order greater or equal than 2 are also ramification points. Those should be included here, or else it should be stated why they are excluded.
p.4: "... has just poles located at ..." would read better as "... only has poles located at ..."
p.4: For the last bullet point for the subclass of symplectic transformations that leave the correlators invariant, it would be useful to add a reference here where this statement is proved.
p.5: eq.(2.4): integration is not really defined here. It is stated below that the "integration constants" will not play any role, but this is more subtle than that if the spectral curve has genus > 0 (the path of integration should be defined as there are non-trivial A and B-cycles). Perhaps it could be stated here that these definitions are only for genus 0 spectral curves?
p.5: "... but x and y interchanged." should be "... but with x and y interchanged."
p.5: "... are dual to each over." should be "... are dual to each other."
p.5: eq.(2.9) same as for eq. (2.4). This focuses on genus 0 spectral curves.
p.5: last paragraph. "... for spectral curves with meromorphic $x,y: \Sigma \to \mathbb{C}$, i.e. there exists an irreducible polynomial $P(x(z), y(z)) = 0$ for $x \in \Sigma$." This is only true if x and y are meromorphic functions on a compact Riemann surface $\Sigma$. This should be stated.
p.6: "... suggested by Gukov and Sulkowski." The relation between topological recursion and quantum curves was proposed studied many years before the paper of Gukov and Sulkowski. It could perhaps be traced back to https://arxiv.org/abs/0901.3273 by Bergere and Eynard.
p.6: "... can also be represented as the vanishing locus of a polynomial..." Again, this is only when x and y are meromorphic functions on a compact Riemann surface $\Sigma$.
p.6: In the equation below, you write the algebraic curve as a subset of $\mathbb{P}^2$. Is it clear that we want to projectivize in $\mathbb{P}^2$ here? And why?
p.6: "... having such an differential operator ..." should be "... having such a differential operator ..."
p.6 The conjecture that there exists a quantisation stated below eq. (2.11) is only expected, as stated, for genus 0 spectral curves. Otherwise, the wave function should be modified (the non-perturbative wave function should be used). This should probably be stated here, or the restriction to genus 0 spectral curves be explicitly mentioned.
p.7: "... is construction again..." should be "... is constructed again..."
p.7: "... that both wave function are related ..." should be "... that both wave functions are related..."
p.7: "... that it Fourier transform is again..." should be "... that its Fourier transform is again..."
p.7: "... has no ramification points implying the ..." should be "... has no ramification points implying that..."
p.7: "... (or Bairy function respectively)..." What does this mean? Respectively to what? There is only one wave function constructed from the dual $\tilde \omega_{g,n}$ here...
p.8: "... since the pullback to the x-space form ..." should be "... since the pullback to the x-space from..."
p.8: "... or equivalently a specific solution of the of the quantum spectral curve." should be "... or equivalently a specific solution of the quantum spectral curve."
p.8: "... is therefore an other way ..." should be "... is therefore another way ..."
p.8: There should be a space before [EGFG+23].
p.8: "An other way is to construct..." should be "Another way is to construct..."
p.8, eq.(2.16): I don't understand the equation. Shouldn't there be a product over i for the first term on the right-hand-side?
p.8, eq.(2.17): The LHS is a function of$x_1$, the RHS of $x$. This should be the same variable.
p.8: "... we will need first to define the the following graphs:" should be "... we will need first to define the following graphs:"
p.9: "We will abuse the notation $x_i = x_i(z_i) = x_i(y_i) = x(z_i)$..." is a little bit confusing to me. I understand the abuse of notation $x_i = x(z_i)$, and perhaps $x_i = x(z_i) = x_i(z_i)$ by extension. But can we really think of $x$ as a function of $y_i$ always? (Same for the reverse statement.) Perhaps just adding a word or two here might help, although this may be deemed unecessary by the author.
p.9: "... by TR by the spectral curve..." should be "... by TR on the spectral curve..."
p.9: "Since both wave function ..." should be "Since both wave functions ..."
p.9: "... a way to generated asymptotically..." should be "... a way to generate asymptotically..."
p.9: "The Fig. 3 shows the ..." should be "Fig. 3 shows the..."
p.10: In Corollary 2.6, it should be stated here that the spectral curve is genus 0 (otherwise you can't set $y(z) = z$ or $1/z$), and also that only simple ramification points are allowed for x (which is a condition under which Theorem 2.5 is proved.)
p.10: (A001187) refers to what?
p.10: "... can indeed by turned into..." should be "... can indeed be turned into..."
p.10: "... get an other explicit formula..." should be "... get another explicit formula..."
p.10: Proposition 2.7: same conditions as Corollary 2.6.
p.11: "This proves the the assertion." should be "This proves the assertion."
p.11: "... are the same as computed by TR." should be "... are the same as computed by TR:"
p.12: Example 2.9. Unless I am mistaken, Theorem 2.5 is only proven for genus 0 spectral curves that are algebraic (i.e. x and y are meromorphic functions on a compact Riemann surface) and such that x and y only have simple ramification points. (This is what is stated in Theorem 2.5.) So the same is true of Corollary 2.6 and Proposition 2.7. However, the r-spin Airy curve has a higher order ramification point (unless r=2, but then it's just the Airy curve), so Proposition 2.7 does not in principle apply. What then justifies applying it in eq. (2.28)? Is it expected that the formula would still hold for curves with higher ramification? If so, this should be discussed. It should at least be mentioned that the application of Proposition 2.7 to this particular case is conjectural, since it does not fall within the class of cases for which the proposition is proved, unless I am mistaken.
p.13: "... which was can be used..." should be "... which can be used..."
p.13: "... in the n=1 to derive..." should be "... in the n=1 case to derive ..."
p.13, Example 2.10. Same comment as for Example 2.9, but here x and y have log singularities. So they are not meromorphic on a compact Riemann surface, i.e. the spectral curve is not algebraic. So, unless I am mistaken, this spectral curve does not fall within the class of curves for which Theorem 2.5 and Proposition 2.7 are proved. This should be commented on and stated.
p.13: In fact, just above eq.(2.33) it is stated that Corollary 2.6 and Proposition 2.7 don't apply, but some other formula is used instead. Where is this formula coming from? This should be explained, or there should be a reference here.
p.13: "... (the explicit definition how these objectds are defined will be ..." should be "... (the explicit definition of these objects will be ..."
p.14: "... for the same curve we exchanged x and y." should be "... for the curve with x and y exchanged."
p.14: "We conclude an obvious mismatch..." should be "We conclude that there is an obvious mismatch..."
p.14, footnote: there is a missing period at the end of the footnote.
p.15: "Integrating once wrt x ..." Why not write "Integrating once with respect to x ..." ?
p.16: "... appeared in expansion of the ..." should be "... appeared in the expansion of the..."
p.16: The statement in the box is the main conjectural statement of the paper. For clarity, I think that it should be stated here that it is conjectural, as it is not proved in the paper. It is however checked in a few examples below.
p.16: "... we find as a consequence of the x-y duality formula, Corollary 2.6 and Proposition 2.7 the explicit form" would read better as "... we find, as a consequence of the x-y duality formula, Corollary 2.6, and Proposition 2.7, the explicit form"
p.17: "... by Gukov and Sulkowki [GS12, (3.20)], which conjectures" may read better as "... by Gukov and Sulkowki [GS12, (3.20)], who conjecture that"
p.17: "... we identify at order $\hbar^{2g-1}$ at the lhs with ... from the rhs under ..." would read better as "... we identity the terms of order $\hbar^{2g-1}$ in the lhs with ... from the rhs, under..."
p.18: "... tells us that both, $W_{g,1}$ and $W^\vee_{g,1}$, should not vanish..." should be "... tells us that both $W_{g,1}$ and $W^\vee_{g,1}$ should not vanish..."
p.19: "... and we conclude the following statement:" would read better as "... and we obtain the following statement:"
p.19: "Next, we discussion the compatibility..." should be "Next, we discuss compatibility ..."
p.19: "This obviously aligns with prediction of Gukov..." should be "This obviously aligns with the prediction of Gukov..."
p.20: "The proposition states that for a given asympotitic ..." should be "The proposition states that for a given asymptotic..."
p.20: "It was show and discussed..." should be "It was shown and discusssed ..."
p.21, first paragraph: What does the author mean by the statement that the logarithmic singularity of x is "cancelled" by the logarithmic singularity of y? Perhaps this statement should be explained or made clearer?
p.21: "... is understood as asymptotic series..." should be "... is understood as an asymptotic series..."
p.22, Remark 4.1. The reference [DLN16] is used for the proof of TR for orbifold Hurwitz numbers. Perhaps a further reference to https://arxiv.org/abs/1301.4871 should also be added, where this statement was also proved independently at the same time.
p.22, eq.(4.6) There should be a period at the end of the equation.
p.23, eq.(4.8) and (4.9) There should be commas at the end of these two equations.
p.24, Remark 4.3: There is a closing bracket ) missing at the end of the paragraph for the spectral curve.
p.25: "... does not fit into the class considered Sec 3.2..." should be "... does not fit into the class considered in Sec 3.2..."
p.25: "... we have for $\tilde \omega^\vee_{g,n} = 0$..." should be "... we have $\tilde \omega^\vee_{g,n} = 0$..."
pp.24-25: The discussion of the t=0 limit and the t\neq 0 case in Section 4.3 is a little confusing to me. What is the author doing precisely in this section? The t=0 curve is fine, x-y duality applies, this was studied previously. However, as stated, for t \neq 0 the curve does not fit within the class studied. But it appears that the author is still using the x-y duality formula (or Proposition 2.7) for t \neq 0 to get eq.(4.18)-(4.20)? If so, why? Is it justified? What are the resulting $\tilde W_{g,n}$, are these supposed to be the same as those obtained from TR on the curve when t \neq 0? This is not obvious because, as stated, the limit t->0 is non-trivial here, as ramification points collide. So while this works for t=0, this doesn't mean that it should work for t \neq 0. I am just a little bit confused here as to what the author is doing, what it is assumed, what is a conjecture, etc. Perhaps this section could be written more clearly?
p.26: "... (considered in by Pandharipande [Pan00, Theorem 2])..." would read better as "... (considered by Pandharipande in [Pan00, Theorem 2])..."
p.26, Example 4.5. At some point in the example, the author states that he chooses a particular order for the contour integrals. It is clear that we can choose any order here? Shouldn't the order fixed already when the Laplace transform of the x-y duality formula is stated (for instance in the equation just above Example 4.4 in p.25)? Presumably, an order is chosen for this statement to be precise, right? Otherwise the statement would be ambigous, since, as argued, the order of the contours matters.
p.26: "... have vanishing contributions except of $2^{n-2}$..." should be "... have vanishing contributions except for $2^{n-2}$..."
p.27: "... will interplay with the resurgence since ..." would perhaps read better as "... will interplay with resurgence since..."
p.27: the last bullet point is not a sentence. Perhaps it could be reformulated?
Recommendation
Publish (easily meets expectations and criteria for this Journal; among top 50%)

---

## Round 2 · Referee Report · Anonymous (Referee 1) · 2024-7-8

Strengths

I can only repeat the points from the initial report:

  1. New innovative idea of deforming the topological recursion in a way that makes it compatible with the x-y duality formula beyond the known algebraic case.

  2. Many new and known formulas reproduced as a direct application of this new idea.

  3. The paper has already enriched the realm of topological recursion and inspired new research in this area.

Weaknesses

None

Report

This paper addresses questions of absolute importance on the edge between integrability, topological string theory, enumerative geometry, and matrix models. It features a new idea that has crucial importance for the field of topological recursion, and this idea immediately generates a huge number of applications; many of them worked out in detail in this paper.

I strongly believe that all the journal criteria are met; moreover, thanks to a revision, the paper was improved qua presentation and qua accuracy of statements. The paper has my strongest recommendation for being accepted.

Requested changes

None

Recommendation

Publish (surpasses expectations and criteria for this Journal; among top 10%)

---

## Round 2 · Referee Report · Anonymous (Referee 2) · 2024-8-9

Report

The author addressed the comments in the initial referee report. Therefore, as explained in that report, I suggest publication.

Recommendation

Publish (easily meets expectations and criteria for this Journal; among top 50%)

---

## Round 2 · Author Response

This revision mostly includes corrections of typos found by the second referee.

---

## Round 2 · List of Changes

All points raised by the second referee are corrected or specified. I will list the corrections that are not straightforward by the referees comments:
p.4: the reference which proves in invariance under $y\to y+R(x)$ is just the https://arxiv.org/abs/math-ph/0702045 Thm 7.1
p.5: I have specified for the integration that it is local, with base point close to $z_i$ and I am also just considering genus zero spectral curves
p.24-25,Sec.4.3: I have added; "We will now discuss that taking this curve and the (conjectured) application of the $x-y$ formula together with the proposition of this article is compatible, also with taking singular limits, for instance, of colliding ramification points. This will further support the proposed structure of Sec. 3.2."
This explains the purpose of Sec. 4.3.
p.26: I have mentioned just before Example 4.4 that the order of integration is relevant, and it has to be the same for each summand in the sum over all $\sigma\in S_n$

Throughout the article: I have tried to make it clearer whenever a formula is conjectured (like for x-y duality and TR for higher order ramification points, or including logarithmic singularities for x,y).

I have added a few footnotes stating that the proposed extension of TR in this article was further developed, extended and proved in https://arxiv.org/abs/2312.16950. These are the footnote numbers: 2,5,7

---

## Editorial Decision

published